# Buried deep freshwater reserves beneath salinity-stressed coastal Bangladesh

Huy Le [1] ✉, Kerry Key [2,3] ✉, Michael S. Steckler [2] ✉, Nafis Sazeed [4], Mark Person[4], Anwar Bhuiyan[5], Mahfuzur R. Khan [5] & Kazi M. Ahmed [5]

Aquifer overexploitation and saltwater intrusion threaten freshwater resources in coastal regions worldwide. In Bangladesh, arsenic contamination further reduces shallow freshwater availability. In the coastal zone, much of the shallow groundwater is saline, while the availability of deeper fresh groundwater remains poorly understood. Here, we utilize deep-sensing magnetotelluric soundings to image contrasts in electrical resistivity between fresh and saline groundwater along the Pusur River in the Ganges-Brahmaputra Delta. Our data reveal two distinct deep freshwater bodies separated by a high-salinity zone. We propose that these aquifers formed during the Last Glacial Maximum sea-level lowstand and are protected by overlying fine-grained sediments, whereas the Ganges paleovalley incision, followed by marine transgression and deposition, created the saline gap. Our work maps potential resources for this water-stressed region and suggests that the interplay between past sea-level cycles, sedimentation, and hydrogeological processes demonstrated here may also control the distribution of fresh groundwater in other deltas.

Freshwater scarcity, exacerbated by climate change, is a severe problem in coastal regions worldwide[1–5]. In coastal Bangladesh, particularly in the southwestern (SW) region, high population density and saline groundwater place substantial pressure on freshwater resources[6,7]. Relative sea-level rise, enhanced by embankments[8] and shrimp farming[9], has intensified saline intrusion and further strained freshwater availability[10,11]. Farther inland, shallow aquifers are often contaminated with arsenic (As) and other pollutants[12–14], further limiting potable and irrigation water. It is crucial to identify sustainable management strategies and new freshwater sources to address this pressing water crisis. Deep groundwater (>150 m) is one of the vital water supplies, as it has low salinity and As levels[15–17]. Multiple studies have suggested the presence of deep groundwater at a regional scale[6,15] and evaluated the susceptibility of deep aquifers to As contamination[18,19], but little is known about their extent and the underlying hydrogeological processes.

Electromagnetic (EM) geophysical methods, sensitive to pore fluid salinity, effectively map resistivity variations in groundwater systems[20–22]. Freshwater-saturated sediments exhibit significantly higher resistivity than their saline counterparts, enabling EM methods to delineate freshwater resources within conductive host environments, such as saltwater-dominated zones in sedimentary basins. While near-surface EM methods have been used to map shallow groundwater and Holocene lithology in other parts of Bangladesh[23,24], these techniques are insufficient to delineate deep targets. To image the deep, large-scale aquifers in coastal Bangladesh, we employed broadband magnetotelluric (MT) soundings, a non-invasive geophysical method capable of sensing electrical resistivity from depths of meters to several kilometers and beyond (see Methods). We collected MT data at 25 sites along riverbanks and fields adjacent to the Pusur River, a Ganges River distributary in SW Bangladesh, creating a 120 km-long MT transect from Khulna City to the coastline at the southern edge of the Sundarbans mangrove forest (Fig. 1c). With an average site spacing of about 5 km, the transect captures the large-scale spatial distribution of fresh and saline groundwater in the study area.

[1]Department of Earth and Environmental Sciences, Columbia University, New York, NY, USA. [2]Lamont-Doherty Earth Observatory, Columbia University, New York, NY, USA. [3]Now at Deep Blue Geophysics, LLC, Los Angeles, CA, USA. [4]New Mexico Institute of Mining and Technology, Socorro, NM, USA. [5]Department of Geology, University of Dhaka, Dhaka, Bangladesh. ✉e-mail: hdl2115@columbia.edu; kkey@deepbluegeophysics.com; steckler@ldeo.columbia.edu

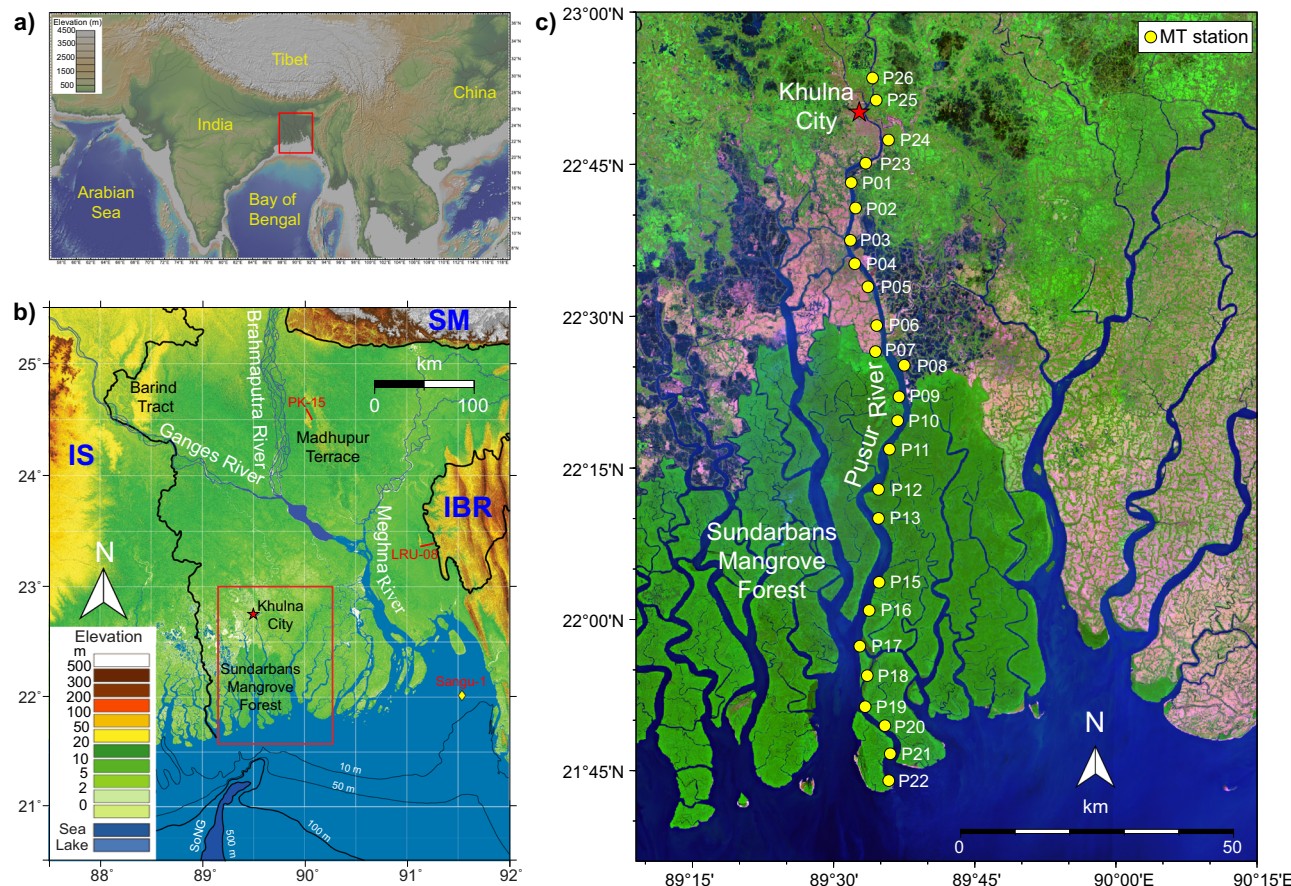

**Fig. 1 | Geological setting and magnetotelluric survey location. a** Map of southern Asia highlighting the study area (red box) within the broader Ganges-Brahmaputra-Meghna (GBM) basin. **b** Topographic map of the GBM basin, showing political borders (thick black lines), subaqueous delta depth contours (thin black lines), and the Swatch of No Ground (SoNG) submarine canyon. Adjacent tectonic provinces of the GBM are the Indian Shield (IS), Shillong Massif (SM), and Indo-Burma Ranges (IBR). Two red lines indicate locations of seismic data used to infer porosity for salinity estimation (see Supplementary Fig. 6). The yellow diamond indicates the offshore Sangu-1 well. The red box outlines the survey location in the southwestern GBD. **c** Land-use satellite map illustrating the survey transect along

the Pusur River, a Ganges distributary. Yellow circles mark the magnetotelluric station locations. Pink and green shading denote fallow agricultural fields and vegetation (including the Sundarbans Mangrove Forest), respectively. Dark blue to purple shading represents brackish-water shrimp farms (Fig. 1a is made with GeoMapApp[72] (www.geomapapp.org) / CC BY; Topography in Fig. 1b is from Shuttle Radar Topography Mission (SRTM). Color-shading basemap in Fig. 1c is a Sentinel-2 Image from March 2nd, 2022, processed by Christopher Small with RGB colors corresponding to SVD (substrate-vegetation-dark)[73]. Figure 1c is made with pyGMT[74].

The Ganges-Brahmaputra Delta (GBD), encompassing over 100,000 km² and formed by the convergence of these two eponymous Himalayan rivers, is the largest delta in the world (Fig. 1). Two-thirds of the GBD lies within Bangladesh, bordered by the Indian Shield to the west, the Shillong Massif to the northeast, and the Indo-Burma Ranges to the east[25]. Annually, the Ganges and Brahmaputra Rivers transport an estimated 1 billion tons of sediment eroded from the Himalayas to the delta[26]. The sediment load, ranging from coarse sand to clayey silt[27,28], consists of medium to coarse sand (28%), very fine to fine sand (38%), and muds (34%)−proportions that are roughly equal within the delta[29]. These sediments are coarsest upstream and progressively fine downstream[30]. Pleistocene terrace uplands, such as the Barind Tract and Madhupur Terrace (Fig. 1b), act as natural barriers controlling sediment dispersal and channel migration[28,31,32]. Meanwhile, the Sundarbans mangrove forest, located in SW GBD, functions as a sediment storage area and a fresh-to-saltwater transition zone for surface water. High sedimentation rates[33,34] drive basin subsidence due to compaction and flexural isostatic loading, increasing toward the coastline, and simultaneously contribute to delta-front seaward progradation[35–38]. This substantial sediment budget is primarily transported during the wet summer monsoon season, when high precipitation and river discharge facilitate sediment delivery to the lower delta plain[39].

Sea-level fluctuations, coupled with sedimentary processes, have been dominant factors controlling the formation of deep aquifers in coastal GBD[40,41]. Marine transgressions and regressions significantly influenced stratigraphy and depositional environments, affecting the emplacement of both fresh and saline water. During the Pleistocene sea-level lowstand, the Bengal delta shoreline regressed 150–200 km offshore[42,43], exposing the continental shelf and facilitating freshwater recharge over vast areas. Conversely, the rapid sea-level rise during the early Holocene caused the tidal zone and shoreline to migrate ~100 km inland[36,40] from their present-day positions. The stronger monsoon during the early Holocene increased river discharge and sediment supply, initiating seaward progradation of the GBD[30,31]. This dynamic interplay between sea-level cycles and sediment supply created conditions conducive to forming deep freshwater aquifers overlain by shallower saline groundwater. Delineating the freshwater distribution and dimensions of these coastal aquifers is a primary objective of this study, as their extent remains largely uncharted.

## Results and Discussion
### Electrical resistivity of deep groundwater systems
The regularized 2D inversion of the MT data (see Methods) yielded a resistivity model characterized by two prominent resistive zones, R1

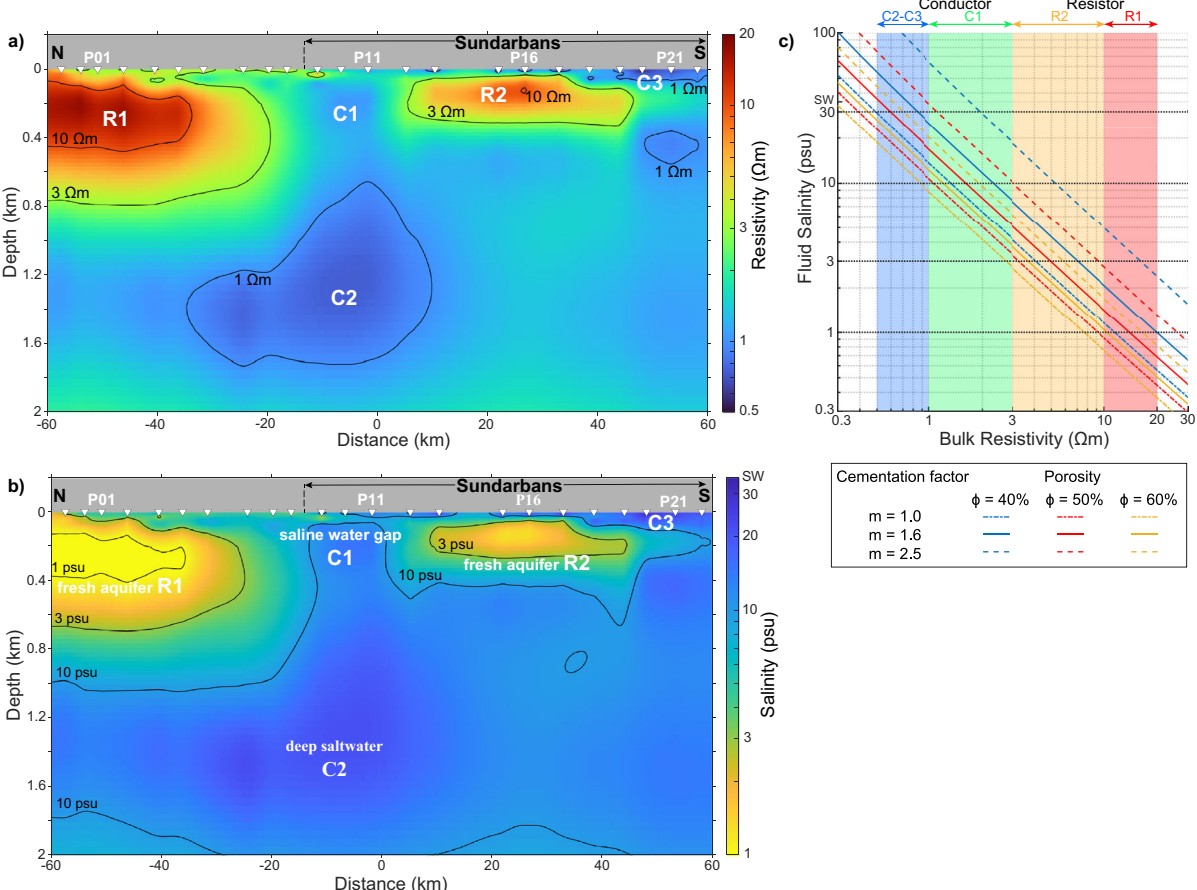

**Fig. 2 | Preferred resistivity and salinity models. a** 2D electrical resistivity model derived from inversion of magnetotelluric data. Yellow/warm colors represent resistive features (up to 20 Ωm), associated with freshwater zones (R1, R2). Blue/cold colors indicate conductive features (0.5–2 Ωm), associated with saline zones (C1, C2, C3). The 1 and 10 Ωm contours delineate the boundaries of conductive and resistive zones, respectively, with the 3 Ωm contour marking the transition between them. **b** The corresponding 2D salinity model estimated using Archie's law and the Practical Salinity Scale 1978 (PSS78; unit: practical salinity unit) (see Methods). Yellow/warm colors represent fresh, low-salinity regions. Blue/cold colors indicate saline, high-salinity regions. The 3 psu contour defines the freshwater boundary,

while the 10 psu contour marks the transition between low- and high-salinity brackish water. The northern freshwater aquifer (R1) and the southern aquifer within the Sundarbans (R2) are separated by a saline water gap (C1). Color scales in (**a**) and (**b**) are logarithmic ($\log_{10}$), but values are labeled in linear units. **c** Pore fluid salinity varies with bulk resistivity, porosity and cementation factors. Salinity is calculated as a function of bulk resistivity ($\rho_b$), porosity ($\phi$), and cementation factor ($m$) by using Archie's law[68] and PSS78[70] with fixed pressure and temperature conditions: P = 0 dbars and T = 30 °C. The color bands denote resistive and conductive zones in Fig. 2a. Porosity and cementation factor values illustrate different lithological conditions of sediments.

and R2, separated by a conductive zone, C1 (Fig. 2a). R1 extends down to 800 m depth along a 40 km span in the northern part of the transect, with a slight southward dip in its upper boundary. R1 is likely to extend farther north beyond the survey area. R2, located within the central Sundarbans, spans 40 km and reaches a depth of 250 m. Notably, R1 exhibits a broader zone of resistivity exceeding 10 Ωm compared to R2, with a maximum resistivity of ~20 Ωm, double that of R2. The transition between resistive and conductive zones is more gradual along the boundary of R1 than that of R2, particularly the lower boundary.

The conductive zone C1, situated in the northern Sundarbans, forms a ~20 km wide discontinuity between resistive zones R1 and R2, potentially connecting vertically with a deeper conductor, C2, extending from 700 to 1800 m depth. A shallower conductive zone, C3, confined within the upper 70 m near the coastline in the southern part of the Sundarbans, exhibits the lowest resistivity (0.55 Ωm) encountered along the entire transect. This shallow layer gradually increases in resistivity northward, away from the coastline.

## Salinity interpretation

We estimated groundwater salinity by converting bulk resistivity to pore fluid salinity, using lithological information from well logs and a model of water conductivity as a function of salinity (see Methods).

The calculations incorporated variables, such as temperature, pressure, sediment porosity, compaction, and cementation factor. Our preferred salinity model (Fig. 2b) defines freshwater (<3 psu) and saltwater (>30 psu) zones, with the 10 psu contour marking the transition between low- and high-salinity brackish water. This model reveals a clear correlation between resistive zones (R1, R2) and low-salinity aquifers, as well as between conductive zones (C1, C2, C3) and high-salinity groundwater.

## Freshwater reserves

Although our results provide first-order details of deep fresh aquifers along the Pusur River, explaining the origin of these aquifers and the factors controlling their distribution using resistivity models alone is challenging. Therefore, we incorporated additional nearby data–including stratigraphic information, as well as groundwater age and salinity measurements from boreholes–to further investigate aquifer characteristics (Fig. 3). Lithological and isotopic data from sediments in boreholes up to 90 m depth in the northern part of our profile[28,30,44] reveal spatial variations of the Holocene-Pleistocene sediment boundary (Fig. 3). Shallow Holocene aquifers are delineated in the upper 90 m[7], whereas the deep aquifers (>100 m) identified in this study are likely associated with Pleistocene or older strata[45].

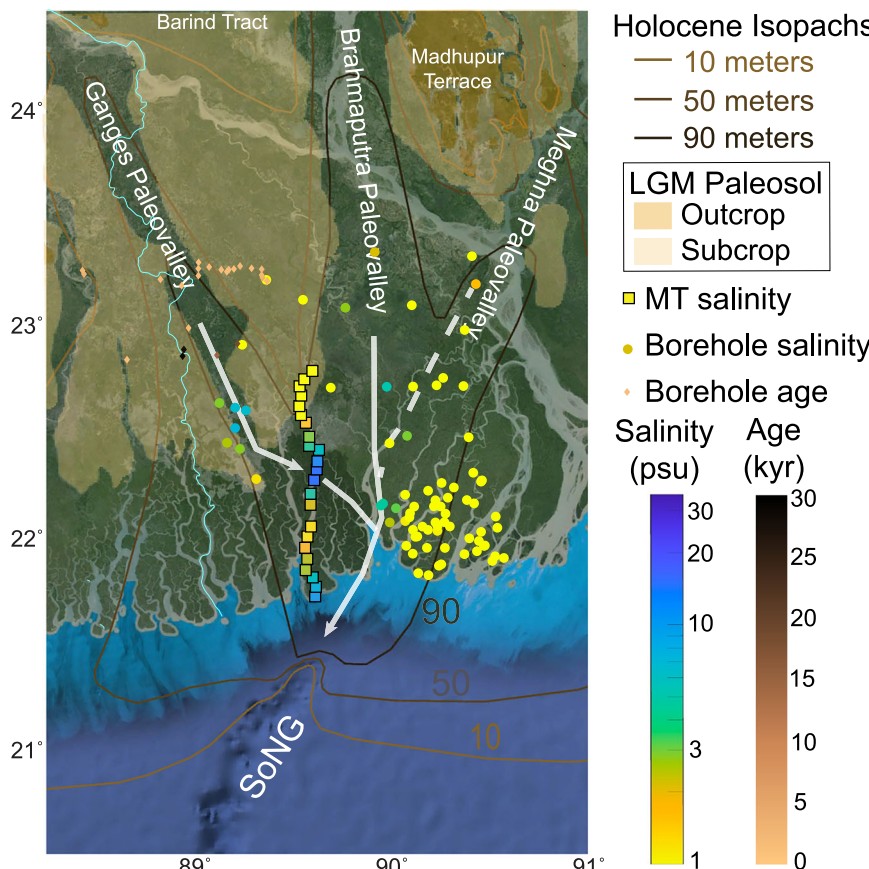

**Fig. 3 | Map showing groundwater salinity and Holocene isopachs in Southwest Bangladesh.** The squares indicate salinity estimated from the magnetotelluric (MT) data at each station, while the circles illustrate groundwater salinity from borehole surveys[13,75–78]. The colors of the squares show the freshest part of the estimated salinity beneath each MT station at approximately 150–250 m depth in Fig. 2b. In contrast, the borehole salinity is converted from electrical conductivity (EC) measurements (unit: μS/cm) at depths ranging between 120 and 350 m using PSS78 at P = 0 dbars and T = 30 °C. The location of the Last Glacial Maximum (LGM) paleosol[42] and Holocene sediment thickness[36] contours provide complementary information on the location of interfluves with weathered soils. The diamonds show groundwater [14]C ages from Khan et al.[13] with depth greater than 150 m within the paleovalley and interfluves areas. The saline water gap in our survey transect is associated with the incised Ganges paleovalley, whereas the deep fresh aquifers underlie the interfluves. Base map image: Google Earth (Data SIO, NOAA, U.S. Navy, NGA, GEBCO; Image Landsat / Copernicus).

We propose that resistive zones R1 and R2 represent primarily freshwater reservoirs formed due to the low sea-level stage in the Last Glacial Maximum (LGM) and subsequently sealed by a deposition of finer-grained, low-permeability layer[44]. During the LGM lowstand, when sea level was as much as 120 m lower than today, and the major rivers incised deep, wide paleochannels across the GBD[32,40,44,46], converging in the lower delta (Figs. 3 & 4). At that time, the shoreline was near the present continental shelf edge[41], exposing the entire shelf to freshwater recharge[45] (Fig. 4a). Weathering of the Pleistocene interfluves exposed during glacial periods (Supplementary Fig. 7) created a stiff clay layer known as the Last Glacial Maximum Paleosol (LGMP)[44]. The subsequent rapid sea-level rise flooded the coastal GBD[45,47], trapping freshwater beneath fine-grained transgressive sediments and the LGMP[31,44,47]. The LGMP acts as a barrier inhibiting vertical mixing between dense saltwater and lighter freshwater below[44,47]. However, marine sediments deposited within paleovalleys dissect the lateral continuity of the LGMP[44,46,47], such as the NW-SE oriented Ganges paleovalley that intersects the MT profile in the northern Sundarbans (Figs. 3 & 4b) and forms conductor C1. Since the LGMP is crucial for protecting fresh groundwater, its lateral discontinuity across the lower delta[43,47–49] strongly influences the distribution of these aquifers.

Our imaging results suggest that resistor R1 is potentially part of a larger deep freshwater system extending tens of kilometers or more to the north, as R1 maintains its full thickness northwards (Fig. 2a). The 10 Ωm contour at the top of R1 (Figs. 2 & 4c) might delineate the LGMP,

which separates the thick resistor R1 from the overlying thin, conductive Holocene sediments that dip southward. The LGMP layer (subcrop in Fig. 3) inhibits local vertical recharge and prevents saltwater infiltration from shallow Holocene aquifers to R1. Therefore, if active recharge to R1 exists, it likely occurs farther north via lateral flow. The alternating layers of fine- and coarse-grained sediments reduce effective vertical hydraulic conductivity, facilitating the regional lateral flow[19]. Due to the minimal topographic relief of coastal Bangladesh and the low hydraulic gradients, natural horizontal recharge to deep aquifers at regional scales would require thousands of years or more[18,19,45,50]. Although the LGMP inhibits vertical recharge, Holocene-Pleistocene groundwater mixing may still occur near the southern edge of freshwater R1, where the LGMP is absent. In our models (Fig. 2), this mixing potentially occurs within the gradual vertical transition zone between R1 and C1 via vertical salt diffusion[51]. In addition to geophysical imaging and mapped stratigraphy, regional isotopic data provide insights into the origin and flow patterns of coastal aquifers in Bangladesh. A decrease in deep groundwater [14]C activity from northern Bangladesh toward the coastline, along with δ[18]O enrichment in deep southern aquifers[52], suggests slow seaward recharge through both vertical and horizontal flow pathways and long residence times of up to ~25 kyrs. Since R1 is likely the southern, distal end of a larger coastal aquifer located below the mapped interfluvial LGMP subcrop[44] (Fig. 3) and lies in an area of low [14]C activity[52], we infer that R1 may partially to mostly contain Pleistocene-age freshwater.

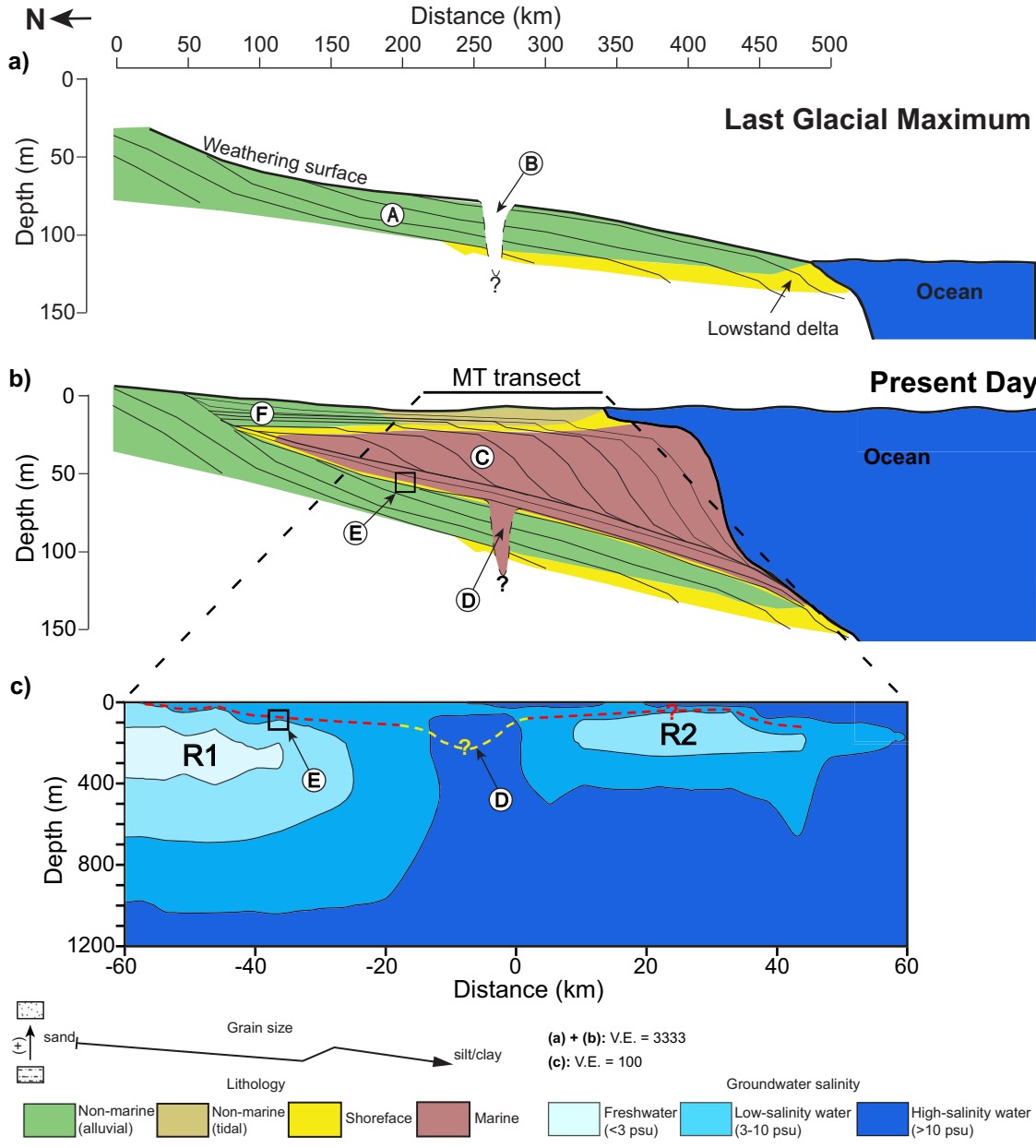

**Fig. 4 | Schematic diagrams illustrating the formation of fresh aquifers in Southwest Bangladesh. a** Last Glacial Maximum (LGM): During the LGM, low sea levels exposed the lower delta, facilitating recharge and the formation of deep freshwater aquifers within coarse-grained alluvial deposits (**A**). The Ganges River incised a northwest-southeast oriented paleovalley (**B**), influencing the spatial extent of the aquifers. **b** Holocene Sea-Level Rise: Subsequent sea-level rise led to marine transgression and conductive marine sediments deposition (**C**), infilling the paleovalley (**D**). Fine-grained, low-permeability sediments and weathered Pleisto-cene clays (**E**) capped the older fluvial strata, protecting the freshwater aquifers from vertical mixing with saltwater. High sedimentation rates caused delta pro-gradation (**F**). **c** Present-Day Hydrogeological Configuration: The distribution of fresh aquifers, inferred from magnetotelluric and stratigraphic models, is shown. Red dashed lines represent the LGM paleosol, while the yellow dashed line indicates the incised paleovalley. Question marks denote uncertainties in the inferred structures. Figure 4a & b only show depths down to 150 m to explain the formation of paleovalley and LGM paleosol distribution, whereas Fig. 4c extends deeper to capture the full dimensions of two fresh aquifers, R1 and R2.

Groundwater $^{14}$C ages, based on dissolved inorganic carbon in the interfluve and paleochannel areas that go across the Ganges paleo-valley located to the northwest of our MT transect (Fig. 3) provide further constraints on the origin and dynamics of deep freshwater reserves[13,16]. Within the Ganges paleovalley area, without the LGMP layer and where the Ganges paleovalley fill is sandy (Fig. 3 transect at 23.2°N), reported uncorrected $^{14}$C ages that cluster around 2.0 kyrs at depths of 150–250 m, indicating the downward flow of Holocene As-contaminated groundwater[13]. In contrast, in the interfluve area asso-ciated with the LGMP farther south where the paleovalley is muddy (south off 23°N in Fig. 3), most groundwater samples have $^{14}$C ages

approximately 10-12 kyrs, with two low-As wells exhibiting Pleistocene ages of 15.1 kyrs at 162 m and 29.6 kyrs at 249 m depth[13] (Fig. 3). The older ages in interfluve areas reflect minimal vertical As-contaminated groundwater transport. We reported the published uncorrected $^{14}$C ages instead of corrected ages because of the high uncertainty in correcting for the initial $^{14}$C content[13], however, these $^{14}$C ages deline-ate a clear spatial pattern in the distribution of deep groundwater ages. The older groundwater in interfluve areas suggests that deep confined systems protected by the LGMP, like the R1 reservoir, are likely to preserve paleowater at least from latest Pleistocene with limited slow regional recharge during the Holocene.

The resistive body R2, located within the Sundarbans mangrove forest, appears to be an isolated aquifer disconnected from the deep regional groundwater flow system due to the incision of the Ganges paleovalley (Figs. 2 & 3). This apparent isolation suggests that R2 are primarily relict Pleistocene groundwater with minimal connection to modern active recharge. The smaller thickness and higher salinity of R2 compared to R1 support this interpretation. An alternative explanation revealed by LGMP and isopach mapping (Fig. 3) is that R2 could be connected to flow paths originating in West Bengal India. In this scenario, R2, similar to R1, would contain paleo-groundwater emplaced prior to the Holocene, potentially connected to distal upland recharge farther west at extremely slow rates due to low topographic gradient. The LGMP surface may end or become deeper than the maximum depth of tube wells at the southern limit of data[44]. In the Sundarban Mangrove Forest, the LGMP cannot be mapped due to the lack of wells. If the LGMP did not develop as far south as R2, it would reduce R2's isolation from saline Holocene groundwater and regional mixing. Additionally, the increasing proportion of clayey silt toward the coastline may lower the bulk resistivity of R2, due to enhanced surface conduction effects in clay mixtures[53]. We neglected the effects of clay content on formation resistivity since saltwater conductivity is the dominant factor.

## High-salinity zones

The 20 km-wide conductive zone C1, representing a saline gap between freshwater aquifers R1 and R2, corresponds with the NW-SE oriented Ganges paleochannel incised through the northern Sundarbans, as indicated by tubewell data[52,54] (Figs. 2 & 3). This paleochannel formed during the Pleistocene lowstand, when the Ganges River carved a deep valley and transported sediments seaward (Fig. 4b). During that lowstand, both the Ganges and Brahmaputra Rivers emptied into the shelf-indenting Swatch of No Ground (SoNG) canyon (Fig. 1b). As sea levels rose in the Holocene, saltwater inundated the paleochannels, depositing marine or estuarine sediments and creating the observed saline gap. Although tubewell data only extend to a depth of 91 m, the paleochannels once connected to the present-day SoNG canyon, itself incised to depths greater than 500 m during the Pleistocene[42] (Figs. 1 & 3). We speculate that the paleochannel could extend up to a few hundred meters deep at our MT transect (Fig. 4b, c), providing abundant accommodation space for saltwater-filled sediments during marine transgression, thereby giving rise to feature C1.

Our data reveal a thin, relatively conductive layer overlying freshwater aquifer R1, which correlates well with the south-dipping Holocene deposits (30 to >90 m thick) documented in tubewell data[29]. Increased sediment supply during the early Holocene, driven by a stronger Indian monsoon, resulted in the progradation of a broad tidal zone over the marine Holocene deposits[27,38,55,56]. Shallow salinity decreases northward from 30 psu at conductor C3 near the coast to 3 psu at the northern end of the transect, 120 km inland, coinciding with the maximum Holocene transgression[40]. This gradient mirrors the high surface water salinity observed during the dry season, extending inland to the topographic break between the fluvial and fluvial-tidal delta[50]. These findings confirm extensive shallow saltwater intrusion within the coastal GBD, contributing to freshwater scarcity during the dry season.

The broad conductive region below two freshwater aquifers, between depths of 1 and 2 km (Fig. 2), could represent marine clay extending across the Bengal Basin. The offshore Sangu-1 well (Fig. 1) yielded sands down to ~600 m depth, overlying sandy shale[57] within the Pleistocene Megasequence 3 (base 0.4–1.9 Ma[58]). Since conductor C2 exhibits resistivity values lower than its surroundings and salinity levels close to those of seawater, it may represent offshore shelf deposits seaward of the coast. Based on seismic data[43], we hypothesize that this deep saline unit formed during marine deposition prior to large-amplitude sea-level cycles that extended the shoreline and deposited coarser-grained sediments near the shelf edge. These processes subsequently facilitated the emplacement of freshwater bodies R1 and R2.

## Sea-level change and the formation of deep freshwater resources

Our MT survey along the Pusur River reveals two distinct freshwater bodies: a seaward-dipping wedge extending to a depth of 600 m in the northern part of the transect, and a smaller, shallower fresh-to-brackish zone located closer to the river mouth. Corroborating evidence, including Holocene sediment thickness, LGMP distribution associated with interfluves and paleovalleys, and nearby groundwater [14]C ages, supports the conclusion that the changes in sea-level cycles and subsequent stratigraphic deposition have governed the formation and distribution of the deep freshwater reserves, R1 and R2. A high-salinity zone between the two freshwater bodies correlates with the Ganges paleovalley, which was incised during the LGM sea-level lowstand and subsequently infilled with conductive muddy sediments during the marine transgression. Our results highlight the critical role of paleo-incised valleys in controlling the spatial distribution of fresh and saline groundwater as they are geological features commonly formed in coastal plains worldwide during periods of low sea-level[51,59,60]. Furthermore, paleovalley areas also serve as potential conduits for saline water intrusion into deeper flow systems, thereby contributing to coastal aquifer salinization. This onshore groundwater system mirrors recently studied offshore systems[61,62], where interactions between sea-level fluctuations, sediment transport, groundwater flow, and geological processes shape freshwater formation and distribution over geologic time scales. Our findings suggest that the two identified aquifers represent notable freshwater reserves requiring millennia to develop. These deep aquifers could serve as critical freshwater sources for saline-affected coastal regions in Bangladesh[6,15]. However, their slow recharge rates necessitate careful integration into long-term water management strategies. Compared to other near-surface EM techniques used in groundwater exploration, MT soundings can image the full depth extent of deep groundwater systems beyond just the water table or saltwater-freshwater interface. Understanding the dimensions of aquifers allows for volumetric estimation, which is essential for sustainable water management. While deep freshwater has been locally exploited in coastal Bangladesh, our MT-derived salinity model, combined with lithological data, provides the framework for understanding the distribution of fresh and saline deep groundwater systems. This framework is valuable for strategically locating future deep extraction wells and determining pumping rates. Similar interactions of sea level cycles and fluvial incision may shape deep fresh aquifers in other coastal deltas and continental margins. as sea-level lowstands were widespread globally during the Pleistocene[58–61].

## Methods

### MT data collection and processing

The MT method is a passive EM geophysical technique that measures the temporal variations of electric and magnetic fields to image subsurface resistivity structures. The primary source fields for MT data at lower frequencies (typically below 1 Hz) originate from the interaction between charged particles from the solar wind with Earth's magnetosphere and ionosphere, generating fluctuating ionospheric electric currents, which in turn induce EM fields that propagate downward through the atmosphere and into the subsurface. At higher frequencies (typically above 1 Hz), global lightning discharges are the dominant source of EM fields for MT measurements. As the Earth's interior is more conductive than the atmosphere, the penetrating EM fields interact with the resistivity structures and induce EM signals that are measured across six decades of frequencies.

In March 2022, we collected broadband MT data at 25 sites along riverbanks and agricultural fields adjacent to the Pusur River, a

distributary of the Ganges River in SW Bangladesh. Our survey extended from Khulna City to the coastline (Fig. 1c). We conducted overnight MT deployments with roughly 5 km spacing, yielding MT responses from $10^{-4}$ to $2 \times 10^3$ s periods. This wide bandwidth allowed for sensitivity at multiple depth ranges. Cultural noise near Khulna City, such as 50 Hz electrical powerlines, brick factories, farming, and livestock, degraded the MT signals at five sites in the northern part of the transect. However, we observed a significant increase in the signal-to-noise ratio at sites within the Sundarbans Mangrove Forest. Approaching the coastline, the MT responses at periods greater than 2 s become noisy, likely due to seawater motional induction noise generated by waves and strong water currents. We used remote referencing and robust processing techniques to estimate broadband MT impedance responses[63]. For most sites, we excluded bad data with large error bars and outliers and set an error floor of 5%. We discarded two northernmost sites (P25 and P26) because of their high noise level and set an error floor of 10% for P01, P02, and P03. These processes generally yielded high-quality MT responses in the period range of $10^{-3}$ to $10^3$ s (Supplementary Fig. 1).

### Electrical resistivity inversion methods

To obtain electrical resistivity models, we applied two complementary inversion techniques to the MT data: conventional regularized inversion and a Bayesian sampling method. The non-linear regularized Occam's inversion approach[64] generates a smooth model that fits the data within a specified tolerance, using a model roughness penalty to minimize the introduction of unnecessary structures. This approach is computationally efficient for identifying prominent resistivity features but does not provide quantitative estimates of model parameter uncertainty. Recognizing that model uncertainty is inherent in geophysical inverse problems due to data noise, model simplification, and computational limitations[65], we also employed a Bayesian sampling method to quantify the range of plausible models that fit the observed data.

For the regularized inversion, we employed MARE2DEM[66], a non-linear regularized Occam's inversion code, to invert $\log_{10}$ apparent resistivity and phase of MT responses into a 2D bulk resistivity model. The inversion domain was parameterized using a dense quadrilateral grid in the upper 2 km to resolve near-surface groundwater systems, transitioning to a sparser grid from 2 to 10 km depth to accommodate potential deep structures influencing the longer-period data. A horizontal-to-vertical roughness penalty ratio of 200 was assigned to reflect both the large length-to-depth ratio (120 km: 10 km) of the survey area and to favor models with laterally continuous features, given the approximately 5 km station spacing, which is much wider than our 0 to 2 km depth of interest for groundwater mapping. To account for the conductive effect of the Bay of Bengal at the southern end of the transect, we incorporated an adjacent ocean region with a fixed resistivity of 0.3 Ωm. Our preferred 2D resistivity model (Fig. 2a) achieved a root-mean-square (RMS) misfit of 1.005 after 13 iterations.

In the Bayesian sampling approach, the $\log_{10}$ resistivity model parameters are treated as random variables. We compute their probability density function (PDF) as a function of depth using a parallel tempering reversible jump Markov chain Monte Carlo algorithm[65,67]. By applying parallel tempering, the MCMC algorithm can more effectively explore the entire model space and avoid getting trapped in the local maxima of the probability distribution, thereby reducing potential biases in the sampling results. Additionally, we treat the model parameters, such as the number of resistivity layers, their thicknesses, and depths, as unknowns and include them in the inversion process. The algorithm applies the concept of Bayes' rule:

$$p(\mathbf{m}|\mathbf{d}) \propto p(\mathbf{d}|\mathbf{m}) \times p(\mathbf{m}) \qquad (1)$$

where $\mathbf{m}$ is a vector of model parameters describing subsurface resistivity structure, and $\mathbf{d}$ is the observed data vector. The p(d|m)

term is the model likelihood, which measures the probability of observing the data given a specific model. The p(m) term is the assumed prior probability distribution of the model parameters and is independent of the observed data, reflecting prior knowledge or assumptions about the subsurface. On the left-hand side, our target is the posterior model probability distribution $p(\mathbf{m}|\mathbf{d})$, which quantifies our updated knowledge about the model parameters after considering the observed data. The accuracy of the posterior distribution estimation relies on the careful selection of the likelihood function and prior distribution.

Supplementary Fig. 2 shows representative Bayesian 1D inversion models at four sites, overlaid with profiles extracted from the 2D regularized inversion results. The agreement between the regularized inversion profile and the high-probability density (PDF) regions in the Bayesian models at each site confirms the interpreted distributions of resistive freshwater and conductive saltwater zones at depth. Furthermore, the region between the 5th and 95th percentiles of the posterior PDF represents the range of resistivity values compatible with the data at a given depth, as 90% of the models in the Bayesian ensemble fall within this range.

Although both Bayesian and regularized inversions resolve three resistivity layers in the upper 2 km, the Bayesian models reveal that conductors are better constrained than resistors, highlighting the inherent higher uncertainty associated with resistors in MT inversion. This observation is consistent with the MT method's sensitivity to the conductivity-thickness product; MT is inherently more sensitive to conductors than resistors and thick resistors than thin ones. Given the significant number of Bayesian models indicating the resistivity values for the fresh aquifer vary from 10 to 500 Ωm, we postulate that the 10 Ωm estimate provides a reliable lower bound for the freshwater layer. Additionally, the Bayesian method offers improved resolution of the transition depth between resistive and conductive zones. While the regularized models show a smooth change, the high probability regions in Bayesian models clearly delineate the possibility of a sharp boundary between these two anomalies (Supplementary Fig. 2).

### Salinity estimation

From our resistivity model, we calculated the pore fluid resistivity $\rho_f$ using the empirical Archie's law[68]:

$$\rho_f = \rho_b \times \phi^m \qquad (2)$$

where $\rho_b$ is the bulk resistivity obtained from our 2D inversion results, $\phi$ is the porosity, and $m$ is the cementation exponent. We estimated porosity as a function of depth $\phi(z)$ using exponentially dependent Athy's law[69]:

$$\phi(z) = \phi_o \times e^{-cz} + \phi_{min} \qquad (3)$$

where $\phi_o + \phi_{min}$ is the initial surface porosity, $\phi_{min}$ is the minimum porosity at depth, and $c$ is the compaction factor. Based on constraints from seismic data (see Supplementary Fig. 6), we set $\phi_o = 40\%$, the minimum porosity $\phi_{min} = 15\%$, corresponding to a surface porosity of 55%, and the compaction factor to 0.4 km$^{-1}$ for the upper 2 km. The cementation exponent is poorly constrained and can vary significantly with lithology. Therefore, we conducted a sensitivity analysis using a range of cementation values from 1.0 to 2.5, aiming to achieve an estimated pore fluid salinity near the end of the profile that is at or less than the seawater value of 35 psu.

We converted pore fluid resistivity to salinity using the Practical Salinity Scale 1978 (PSS-78)[70]. The PSS-78 is a dimensionless scale that defines salinity based on the electrical conductivity of seawater, incorporating the effects of temperature and pressure. This approach is crucial for accurately interpreting geophysical data in marine and coastal environments. Considering surface water temperature varies

between 25 °C and 28 °C in tropical regions, we incorporated a geothermal gradient of 28.5 °C/km[71] in the conversion to account for increased groundwater temperature with depth (Supplementary Fig. 5). Since temperature affects fluid conductivity, and porosity governs the volume of groundwater, these two factors are crucial for salinity estimation (Supplementary Figs. 3 and 4). In our study area, the clay content is significantly lower than the proportion of sand and silt sediments, particularly for the deeper Holocene. Moreover, since the conductivity of saltwater is significantly higher than the conductivity of clay minerals; therefore, we can safely neglect clay conduction effects when applying Archie's law to estimate fluid resistivity.

While the regularized 2D inversions used for salinity estimation are inherently non-unique, our Bayesian models provided valuable insights into the uncertainty associated with these salinity models. It is well-known that MT data are unable to resolve the magnitude of highly thin resistive layers due to response saturation. For example, the Bayesian 95th percentiles run up close to the prior upper bound of 1000 Ωm, whereas the 5th percentiles are more clearly resolved by MT data (Supplementary Fig. 2) as they are well above the prior low bound. Therefore, the 10 Ωm value, corresponding to the 3 psu contour in the salinity model, represents the lower bound of the resistivity PDF for the freshwater zones in the Bayesian models (Supplementary Fig. 2). Given the inverse relationship between resistivity and salinity, this suggests that the actual salinity of the fresh aquifers could be even lower than 1 psu—the threshold commonly used to define potable freshwater in hydrology. Therefore, we used a combination of 10 Ωm and 3 psu contours as the bounded values to delineate the freshwater zones. Although further validation with deep boreholes and groundwater samples is needed, our estimated salinity model provides a robust initial picture of onshore deep fresh and saline groundwater distribution in coastal Bangladesh. This highlights the role of glacial sea-level fluctuations in emplacing and preserving deep fresh aquifers in coastal zones, opening avenues for further investigation.

## Data availability
The MT data used in this study are available at https://zenodo.org/records/17353663 and as source data provided with this paper.

## Code availability
The MARE2DEM code used to invert the data is available at http://mare2dem.bitbucket.io.

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

## Acknowledgements

We thank the captain and the crew of the M/V *Kokilmoni*, guide Khaliqul Islam Romi Khaliqul, Bachchu Nazrul Islam of Pugmark Tours and Travel and additional cruise team participants: Dhaka University graduate students Md. Masud Rana, Md. Shahadat Hossain Biplab, Abu Saeed Arman, Mohtasim Shahajad Shohan, Md. Montasir Arafat Akif and Sanju Singha; CNRS scientist Jo Céline Grall from LIENSs, La Rochelle, France; New Mexico Tech graduate student Adrien Camille; and Barishal University professor Md. Hasnat Jaman and graduate student Mohidar Hossain. We thank the forest guards for protecting the team from Bengal tigers in the Sundarbans National Forest. Christopher Small made the remote sensing images that helped locate field sites. Steve Goodbred and Holly Michael provided comments on Holocene stratigraphy and groundwater. The MT survey was supported by the National Science Foundation awards EAR-1925821 to M.S. and EAR-1925974 to M.P.

## Author contributions

M.S., K.K., M.P., and H.L. designed the experiment. H.L., K.K., M.S., N.S., M.P., and A.B. collected the data. H.L. and K.K. processed and modelled the MT data. M.S. modeled the seismic data and interpreted the stratigraphy. M.P. modelled formation pressure. M.K. and K.A. contributed the groundwater sampling data and interpretations. H.L., K.K., M.S., N.S., M.P., M.K. and A.B. contributed to writing the manuscript.

## Competing interests

The authors declare no competing interests.
