## [Transparent Peer Review file · Nature Communications]

Buried deep freshwater reserves beneath salinity-stressed coastal Bangladesh

Corresponding Author: Mr Huy Le

Version 0:

Reviewer comments:

Reviewer #1

(Remarks to the Author)

This manuscript by Le et al. presents the results of an interesting study to delineate deep, freshwater zones in the subsurface in Bangladesh. There is a desperate need for new sources of freshwater for the population as the easier to access to groundwater is from sources contaminated with arsenic and/or affected by increasing salinity. The use of magnetotelluric soundings is a good for identifying broad areas of resistivity contrasts in the subsurface to depths much greater than near-surface methods. Therefore, it is an excellent method for identifying larger bodies of freshwater with relatively minimal effort of data collection over a relatively large area.

This investigation was successful in identifying 2 zones of freshwater extending to depths of 600 m surrounded by regions of saline water. The origin and location of these two zones were placed in the geological context of the Ganges Delta region, relating to the influence of sea-level changes during the glacial and post-glacial periods. Therefore, the freshwater zones are interpreted as being Pleistocene in age. The success was due in part to the processing of the MT data and using methods that could significantly reduce error to allow more confidence in processing results, which therefore strengthens the interpretations.

The success of this study has several implications. First, they were able to demonstrate that the use of MT can identify large areas of freshwater in the subsurface in an area that has desperate needs. The two identified zones have potential as sources of freshwater for decades or more, pending proper water management. Secondly, they demonstrated that MT could be an important method for similar studies in other coastal aquifers throughout the world, not only identifying locations of freshwater but also for improving interpretations of the geologic history of an area as the presence of freshwater zones needs to be accounted for by geologic processes.

I find the extended data/tables/graphs very helpful.

I do have some concerns. The age of the northern deeper freshwater zone are taken from ^{14}C data collected from a depth of 90 m, while these zones extend to depth of nearly 600 m. Also, uncorrected ages were reported which can sometimes lead to significant differences once corrected. However, the 30,000 yr age could put a lower limit to the age. Though not available, an estimate of age of the seaward zone would be beneficial. I would like to see a bit more on the methodology of data collection and the instrument that was used to collect data. This would allow the reader to evaluate if the methodology/equipment are appropriate for this purpose.

Here are some specific comments:

In line 28, embankments are mentioned. It might be nice to be a bit more specific as to what type, even if it is in the reference.

Around line 43, it would be helpful to refer to figure 1.

In a few places, word deep is used. This is a relative term and it is used to describe both the 90 m depth upper aquifer and the water zones that extend to 600 m depth. R1 and R2 are referred to as deep and so is the depth of 90 m that the ^{14}C ages were determined.

In Figure 1b, the location of the seismic lines are very difficult to see, as well as the legend. I am not sure this figure will be larger if published.

Figures 2 are also difficult to read, especially 2c. Personally, I would have liked to see in the caption that the distances are relative to the location of P11.

I am not sure where Table 1 applies.

Reviewer #2

(Remarks to the Author)

The manuscript addresses deep groundwater resources in the Ganges-Brahmaputra Delta, Bangladesh, applying magnetotelluric (MT) soundings to delineate distribution and dimensions of deep fresh and saline groundwater zones in this coastal system. The authors recorded data at 25 sites along a 120 km transect adjacent to the Pursur River in southwestern Bangladesh with a vertical extension in a km-scale and identified two distinct deep freshwater bodies, which they attribute to being remnants of preserved Pleistocene groundwater. The authors base their findings on deposition environments /sedimentary processes and pleistocene sea-level fluctuations forming the deep groundwater systems and recorded electrical resistivities of groundwater.

The manuscript appears technically sound and is very well-written. It applies established up-to-date methods. The topic is of scientific interest and well-worth publishing. However, the authors should make the global relevance of their shown findings more clear and identify the their study's unique characteristics in order to distinguish it better from pure regional applicability. In addition, the authors should highlight the potential of this study relating to implications on other (global/overregional) findings and set this into a broader context.

L27: add more recent references

L29-31: provide references

L46/47: stress significance (->large scale)

L 48-62: This section is rather detailed compared to the subsequent section and may be condensed, (and some hydrogeological aspects are suggested to be included)

L 64: Define deep in your context

L72/73: As indicated in the main comments, this is one point that could be used for highlighting the relevance of the presented study

L93: provide short information why it is the preferred salinity model

L93: define why 3 psu is chosen (and not 1)

L98: specify which isotopes or at least state if sediment or water is meant here

L98/99: In Fig. 3 150-250m are given (instead of: 90m), or is another figure meant here?

L101-103: connection to onshore aquifers?

L12: Fig. 4 a and b cover depths up to 150 m, R1 and R2 fresh water extend (much) deeper and only the topmost parts overlap Fig 4a and b, this should be mentioned more clearly when deriving analogies and processes

L112: why are the deductions made from uncorrected Radiocarbon data? (especially as this (published) data is used as a strong argument here for fossil water in the investigated system)

L106-118 preserved Pleistocene groundwater vs. resemblance to groundwater system offshore New Jersey: It should be clarified, if only the morphology or the (emplacement) processes are expected to be similar

L127: refer to Fig. 2

L152: Sea-Level Change

L302: refer to Fig. 1c

L456: add link

Figures and Tables:

Are Figures 4a and 4c and Table 1 mentioned in the manuscript text?

L 470: please delete this line

Fig. 1: legends in a and b are too small, an in b the red entries are difficult to see

L475: not clear how subaqueous delta depth an SoNG can be distinguished

L481 not clear where shrimp farms are (dark blue to purple appears a bit masked)

Fig. 2: The axis gives distances (+60 km), where P11 appears to be the center. Why did you choose this scale (and the center at P11) instead of 0 to 120 km? And why are Sundarbans explicitly mentioned here?

Fig. 3: connection between Fig. 3 and Fig. 4 somewhat difficult

Fig. 4: maybe insert R11 and R2 into Fig. 4c

Reviewer #3

(Remarks to the Author)

Thank you for the opportunity to review your manuscript titled "Deep fossil freshwater in coastal Bangladesh". The authors have a fascinating dataset with their N-S MT transect across a portion of the southern Ganges delta. Admittedly, the geophysics underlying it are outside my field of expertise and I accepted the profiles derived from the MT data as correct during my review. The authors demonstrate two zones of low resistance groundwater which is justifiably interpreted as freshwater at depths up to 600m-800m (R1) and up to 200m-300m (R2). Utilizing data from other studies in the Ganges delta and further afield, they construct a conceptual model to explain their findings. Unfortunately, the conceptual model is plausible but not well-supported with the data as presented. In particular:

(1) The carbon-14 data used to support the conceptual model lacks geographic specificity, which leads to apparent overinterpretation of the data. The data comes from one study (ref #5, Kahn et al. 2019). Most of the carbon-14 ages in Kahn are Holocene and the subset that are Pleistocene are not identified and located relative to the current study area. Some of the points in Kahn are more than 100km from the closest boundary of the study area. Extrapolating over such a distance is unreasonable in most contexts and certainly in a major delta complex such as the Ganges. Further, the authors argue for the water in R1 and R2 to have been emplaced during the last glacial maximum -- only 7 of the carbon-14 dates in Kahn fall in the range of the LGM, which makes understanding the location of Kahn's data relative to the current study even more crucial.

Confidence in Pleistocene ages for the vast majority of Kahn's samples nearest to the subject study area is key. If they're Holocene, the entire conceptual model falls apart. To invoke Pleistocene emplacement of R1 and/or R2, the authors must unequivocally demonstrate the presence of Pleistocene water in close proximity to their transect.

(2) In the development of their conceptual model, the authors argue that the fresh water identified in R1 and R2 is relict, connate water emplaced during the LGM, as opposed to the terminal portion of an active groundwater system. At several times they contradict this conceptualization by referencing upgradient connectivity and making comparisons to the Ganges system to an offshore system in New Jersey (specifics noted in comments in PDF file). The contradictory statements in the text undermine their conceptual model. Further, the northern extent of the R1 freshwater region extends an unknown distance outside the study area. There is no evidence presented for the reader not to envision this lobe of freshwater continuing upgradient as part of a continuous, active groundwater flow system.

The deficiencies in this study principally stem from its heavy reliance on data from existing published work in areas that do not exactly overlap with theirs to support their interpretations of their interesting MT transect results. Without site specific data to support their interpretations, the authors often find themselves at odds with the data and with their proposed conceptual model. The original work (the MT transect) appears sound and compelling; its interpretation much less so.

Version 1:

Reviewer comments:

Reviewer #2

(Remarks to the Author)

I thank the authors for thoroughly revising the manuscript. It has significantly improved, and many aspects have been clarified. Also, comments have been addressed in detail. Yet, though in discussions absolute age derivations have been largely changed to relative timing where direct constraints were missing, regarding uncorrected 14C ages I still would suggest to incorporate some (or a concise version) of the explanation into the manuscript or supplement that the authors provided in their reply to Comment 13 (reviewer 2) and "Comment 15 of the Direct Responses to Comments in the manuscript", especially regarding the Khan et al. study. Also, highlighting the uniqueness of their study as suggested in the first revision was moderately addressed, which might still be slightly improved. As a minor point I suggest to check for spelling/typos (especially the revised passages).

Reviewer #3

(Remarks to the Author)

This is a review of the revised manuscript titled "Deep fossil freshwater in coastal Bangladesh" by Huy Le and others. The revised manuscript largely addressed comments I had and those of the additional reviewers. I appreciate the authors' improvements to the manuscript, in particular the "Fossil Fresh Aquifers" section. This section provides better background for someone not familiar with the impressive body of work on the GBD and more cohesive support for the authors' interpretations (despite the few remaining concerns about the emplacement mechanism and age of R1 discussed below).

I reread and reviewed the entire manuscript for technical content but stopped commenting on editorial issues after the first 50 lines or so. There were several instances of either awkward phrasing or missing words. An editorial review prior to resubmission is recommended. The following items were noted for consideration by the authors.

L22

consider adding "the" before "location"

L34-35

consider "...deep groundwater at a regional scale..."

L35

consider changing "sustainability" to "susceptibility"

L69-70

The topic sentence on L64 limits important sea-level fluctuations to the Pleistocene

L108-193

I can accept R2 being relict Pleistocene water as it appears to be cut off from the modern flow system by the upgradient paleochannel and is more saline than R1. I have more difficulty with the certainty expressed in the age of R1 and its origin ascribed to a body of water "formed" during the Pleistocene. I think the issue is largely semantics.

Firstly, the origin of the water. The language used in the text implies connate water, although that term is no longer used in the manuscript, e.g:

"freshwater bodies formed during the Last Glacial Maximum's (LGM) sea-level lowstand and subsequently sealed by a finer-grained, low-permeability layer" [L108-110]

"remnants of preserved Pleistocene groundwater" [L180]

Intentionally or not, the language implies the water in R1 originated in that location and has remained there for 12,000+ years. Perhaps it has, however, that has not been demonstrated nor is it not supported by subsequent statements that indicate R1 is the distal end of a larger upgradient flow system (L123-125) with slow transport times (L123-125).

Secondly, the timing of recharge of the groundwater in R1. The location of the 14C ages supporting the assertion of Pleistocene water is provided only as "northwest of our profile". Other than the fact those samples are somewhere in the western portion of the 100,000 km² GBD (L50), the reader has no information whether these data are relevant -- were they collected 1 km, 10 km, or 100 km away? There is a significantly stronger case for transferability if they're nearby. Consider showing the locations on Figure 3 if they are within the map extent. The transport time of "thousands of years or more" (L124-125) is consistent with Holocene as well as Pleistocene recharge. The 14C data presented provide evidence for shallow-deep difference in age in another part of the GBD system having similar characteristics to the region of R1, which is useful, but not conclusive given the lack of information about their location. The evidence presented for Pleistocene water in R1 is not sufficiently compelling to make a definitive assertion as to its age, particularly in light of the wide range and complex distribution of groundwater ages presented by Kahn et al. (2019), from which the example 14C data were taken.

Consider using the uncertainties surrounding the emplacement mechanism and timing of R1 in your favor instead of trying to force certainty with imperfect data. For example, if R1 is the distal end of a long, slow flow system, it is potentially a more usable and renewable resource than connate Pleistocene water. If the age of R1 is mid-Holocene (for example) rather than Pleistocene, it could indicate a more useful/renewable source. Since the groundwater system hosting the freshwater in R1 is relatively coarse-grained (capped by the clayey paleosol), could you pump water from R1 without substantial drawdown? Could you potentially induce a bit more recharge from modern recharge zones or set up managed aquifer recharge systems? The MT transect is a really useful first step to asking the next set of important questions relating to development and management of the freshwater resource in the GBD.

Version 2:

Reviewer comments:

Reviewer #2

(Remarks to the Author)

I thank the authors for their revisions. My comments have been largely addressed. An aspect that is not clear (and which relates to my former review suggestions regarding 14C) can still be found in lines 150 to 152, where the authors state that „Although uncorrected 14C ages are reported instead of corrected ages, which decrease the uncorrected values by 1.0 14C kyr, due to the high uncertainty associated with initial 14C content corrections, ..“ This reads as if all uncorrected ages can be corrected by 1.0 kyr, which does not appear to be the case. I assume the authors intended to state something else (– and certainly not relate to all samples). And if the „1.0 kyr“ is meant to relate to specific samples/locations, some more information or context would be helpful (and it would be interesting to know why corrected 14C can be given in this case and not elsewhere in the study area, in case this is meant here). I would hence would suggest to rephrase this sentence, and also to cut this (5 lines) long sentence into separate sentences.

Furthermore, I would suggest to clarify line 163 where „... the LGMP surface is not mapped“, whether it does not extend as far, or if it could not be mapped (for technical reasons), as in the current form the relation to „...reducing ist isolation..“ (L162) does not appear straightforward.

Reviewer #3

(Remarks to the Author)

The authors have provided an excellent revision to their manuscript. The arguments and conclusions are considerably stronger and make for a compelling interpretation of their excellent MT transect. I offer only a handful of very minor

suggestions for clarification. Best wishes for successful publication!

L81 A reference to Figure 2b would be helpful after "...southward dip in its upper boundary." I found myself looking at figure 2a and trying to understand how the authors arrived at an aquifer depth of 600 m, which is clearly shown in Figure 2b. Alternatively, if it is desired to stay with the resistivity measurements, consider a different depth reference that coincides with one of the contour intervals on Figure 2a.

L113-117 It seems as if two different mechanisms are provided to explain the capping of the freshwater sand units. The first sentence invokes rapid deposition during transgression, while the second sentence invokes a paleosol. Please clarify.

L554 No location provided for data

Date: February 20, 2025

Manuscript Number: **NCOMMS-24-67705**

Title of Article: **Deep fossil freshwater in coastal Bangladesh**

Name of the Corresponding Author: Huy Le

Email Address of the Corresponding Author: hdl2115@columbia.edu

Dear Editor,

We thank all three reviewers for their valuable time and insightful feedback that has helped improve the manuscript. We have incorporated their suggestions into the main text of our revised manuscript. Below we provide the reviewer comments in black text and our replies in blue. We have also included responses to review feedback that was given in the manuscript PDF.

Best regards,

Huy Le and coauthors

Specific Responses:

Response to Reviewer #1:

Comment 1: *In line 28, embankments are mentioned. It might be nice to be a bit more specific as to what type, even if it is in the reference.*

Reply 1: The embankments mentioned in the text are river embankments. We did not extend the context of the embankments further since the main idea of this section is introducing water problems in Bangladesh.

Comment 2: *Around line 43, it would be helpful to refer to figure 1.*

Reply 2: We added "(Fig. 1)" in line 45 to clarify the survey profile with figure 1.

Comment 3: *In a few places, word deep is used. This is a relative term and it is used to describe both the 90 m depth upper aquifer and the water zones that extend to 600 m depth. R1 and R2 are referred to as deep and so is the depth of 90 m that the 14C ages were determined.*

Reply 3: The word "deep" is used in the main text to separate the MT-imaged aquifers from shallow Holocene aquifer. The deep Pleistocene fresh groundwater in our study area is primarily protected by the Last Glacial Maximum Paleosol (LGMP), whose spatial distribution and thickness vary across the delta (Fig. 3). Ravenscroft et al. (2018) Sarker et al. (2021) and other groundwater studies in Bangladesh defined deep groundwater systems differently. Previously, we used the word "deep" and did not specify the fixed value to consider the complexity of variations of groundwater systems. In the revised edition, We added ">100 m" on line 106 to clarify the depth extension of deep aquifers. Note that the upper part of R2 is shallower than 100 m and might not fit to the defined depth range.

Comment 4: *In Figure 1b, the location of the seismic lines are very difficult to see, as well as the legend, I am not sure this figure will be larger if published.*

Point-by-point response to referees' comments

Reply 4: The resolution in the word document is not great. We hope the resolution in the published version is better to see the seismic lines. The legend shows elevation in meters in color scale.

We enlarged the legends for Fig. 1a and b.

Comment 5: *Figures 2 are also difficult to read, especially 2c. Personally, I would have liked to see in the caption that the distances are relative to the location of P11.*

Reply 5: Again, we hope the resolution gets better in the published version. Another way to improve the readability of figure 2 is rotating the figure 90 degrees and enlarging Fig. 2c. Fig. 2c is important because it links bulk resistivity in Fig. 2a with salinity in Fig. 2b via temperature, cementation factor, and porosity. The zero marker of the distance is chosen to be the center of the profile at the beginning of the inversion. It will add unnecessary workload to rescale the distances relative to the location of P11.

We moved the legend of Fig. 2c to the space below it.

Comment 6: *I am not sure where Table 1 applies*

Reply 5: Table 1 went with Fig. 2 because it lists the bulk resistivity and estimated salinity of features in Fig. 2a and 2b. Although, the color scales in Fig. 2a and 2b indicate clearly the values. Since the table is redundant, we removed it from the revised manuscript.

Response to Reviewer #2:

Comment 1: *L27: add more recent references*

Reply 1: We added He, C. et al. (2021). We keep the number of citations close to 50 references to match Nature Communications suggested limit.

Comment 2: *L29-31: provide references*

Reply 2: We added van Geen et al. (2008) and Rahman et al. (2020) papers.

Comment 3: *L46/47: (-> large scale)*

Reply 3: L45-47: "With an average site spacing of about 5 km, the profile captures the *large-scale* spatial distribution of fresh and saline *groundwater* in the study area."

Comment 4: *L48-62: This section is rather detailed compared to the subsequent section and may be condensed, (and some hydrological aspects are suggested to be included)*

Reply 4: This paragraph provides geological and lithological background relevant to groundwater system. Lithology, as shown later in the text, controls freshwater distribution. Adding hydrological background may create duplication in the discussion, such as slow horizontal recharge due to low topography and vertical mixing prohibited due to LGMP.

We added hydrology background in the first paragraph.

Comment 5: *L64: Define deep in your context*

Reply 5: We defined deep in *Fossil fresh aquifers* section on line 104-107. We added more details of hydrology background in the first paragraph.

Comment 6: *L72/73: As indicated in the main comments, this is one point that could be used for highlighting the relevance of the presented study*

Point-by-point response to referees' comments

Reply 6: We did not see any comments relevant to line 72/73 in the main text. We agree that the point of the sentence on line 72/73 is the goal of our study that we emphasized on the first and last paragraphs of the main text.

Comment 7: *L93: provide short information why it is the preferred salinity model*

Reply 7: In geophysics, the non-uniqueness in inverse problems is well-known, in which an infinite number of models can fit a specific dataset. Thus, we applied regularized and Bayesian inversion methods to quantitatively estimate the non-uniqueness in the resistivity models. Thus, the estimated salinity derived from the resistivity inherits non-uniqueness from bulk resistivity models. The second reason is that we used a range of values for porosity, temperature, and cementation factor to estimate salinity and only showed a model with one set of variables which we think are the best available constraints. Therefore, we used the term “preferred resistivity and salinity models” in the main text. We provided variations of these variables as extended data figures in the Supplementary materials.

Comment 8: *L93: define why 3 psu is chosen (and not 1)*

Reply 8: on lines 473-480, we explained why 3 psu is chosen. We used a combination of 10 Ωm and 3 psu as references to delineate freshwater zones. Converting bulk resistivity to salinity by using Archie's law (1942) involves temperature, pressure, porosity, and cementation factor. These variables can vary in practice, so we only provide the estimation of salinity from our resistivity models.

Also, we notice that MT data are unable to resolve the magnitude of highly resistive layers due to response saturation. This is reflected where the Bayesian 95th percentiles run up close to the prior upper bound of 1000 Ωm (Extended Data Figure 2). Thus, MT responses might underestimate the true resistivity of the freshwater bodies, and the actual salinity of the fresh aquifers can be lower than 1 psu, which is the threshold commonly used for drinking water. 3 psu contour in the salinity model is the upper limit of the estimated salinity of fresh groundwater.

We added more details to that discussion on lines 443-453 to clarify our point.

Comment 9: *L98: specify which isotopes or at least state if sediment or water is meant here*

Reply 9: We clarified in the text that the isotopic data is for sediments.

For example, Goodbred et al. (2014) used $^{87}\text{Sr}/^{86}\text{Sr}$, radiocarbon ages, and other chemical signatures to characterize the origin and depositional timing of sediments.

We only used the information from the cited papers to show the spatial variations of the Holocene-Pleistocene sediment boundary.

Comment 10: *L98/99: In Fig. 3 150-250m are given (instead of: 90m), or is another figure meant here?*

Reply 10: on lines 98/99, we referred to hand tubewells data for lithology. 90 meters is the deepest depth that the hand tubewells extend down to. The information from tubewells allows mapping Holocene-Pleistocene sediment boundary, the LGMP, and Holocene sediment thickness. In Fig. 3, we used contours of 10-50-90 meters to depict Holocene sediment thickness and show LGMP distribution as orange colors. The given depth ranges from 150 to 250 m are associated with the freshest part or lowest salinity at our MT sites (the square symbol).

In Figure 3, we added the contour symbol to the legend to clarify Holocene sediment thickness and related the 150-250 m depth ranges with salinity information to Fig. 2b.

Comment 11: *L101-103: connection to onshore aquifers?*

Reply 11: Resistor R1 and R2 are onshore aquifers located within the low-lying coastal GBD. They are not offshore aquifers. We revised in the main text to clarify that R1 and R2 are potentially connected to groundwater recharge areas further north (Pleistocene terrace) and the West Bengal, respectively.

Although R1 and R2 can connect to upgradient recharge systems, the extremely low topography in coastal GBD and low hydraulic gradients requires long timescales for natural horizontal recharge. The LGMP protects Pleistocene groundwater from vertical mixing with shallow Holocene groundwater. The modern vertical recharge can happen where LGMP is terminated, or at the boundary between paleovalley and the interfluvium. Therefore, we proposed that R1 and R2 are primarily Pleistocene fresh groundwater.

Please see the revised “Fossil fresh aquifers” section in the main text for more information.

Comment 12: *L12: Fig. 4 a and b cover depths up to 150 m, R1 and R2 fresh water extend (much) deeper and only the topmost parts overlap Fig 4a and b, this should be mentioned more clearly when deriving analogies and processes*

Reply 12: We agree with the reviewer. The Fig. 4a&b only show depth down to 150 m to explain the formation of paleovalley and LGMP distribution. Thus, we did not extend further down to the bottom of R1 and R2.

We added this clarification to the end of the Fig. 4 caption.

“Note that Fig. 4a&b only shows depths down to 150 m to explain the formation of paleovalley and LGMP distribution, whereas Fig. 4c extends deeper to capture the full dimensions of two fresh aquifers, R1 and R2.”

Comment 13: *L112: why are the deductions made from uncorrected Radiocarbon data? (especially as this (published) data is used as a strong argument here for fossil water in the investigated system)*

Reply 13: In Khan et al. (2019), there are two deep groundwater ages: As-affected areas and low-As areas. For the As-affected areas, the younger age of 2.0 +/- 0.6 kyrs might indicate the downward flow of Holocene As-contaminated groundwater rather than the true age of deep groundwater (>150 m). The As-affected areas do not have the LGMP to prevent vertical mixing and are associated with the Ganges paleovalley areas. For the low-As areas, the older age of 12.0 +/- 4.0 kyrs reflects no or less As contamination. The deep groundwater in these areas is protected by LGMP from vertical mixing or downward flow and is associated with the interfluvium areas. Therefore, they showed groundwater ages of water samples similar to aquifers in our study areas. Although the location of their study is not collocated with our survey area, the two transects in their study go across the Ganges paleovalley located to the northwest of our MT survey.

In the supplementary materials of Khan et al. (2019), they have a section to explain why they did not use the ¹⁴C age correction. The correction applied for the uncorrected ages would reduce the age by about 1.0

^{14}C kyr, but they would have high uncertainty. Table S1 in Supplementary Materials of Khan et al. (2019) have the corrected ^{14}C age of groundwater for the southern transect.

*“ **^{14}C Age correction:** Converting radiocarbon ages to calendar ages is notoriously difficult for DIC in groundwater because several poorly constrained corrections can subtract hundreds to a few thousand years [Geyh, 2000]. One method is to infer an initial radiocarbon age from the youngest sample in a data set that does not contain any detectable bomb-produced ^3H and therefore no bomb-produced ^{14}C either [Hoque et al., 2012, Mihajlov et al., 2016]. Using existing dataset, this would amount to a decrease in the uncorrected ages by about 1.0 ^{14}C kyr (Supporting Figure S2). Additional corrections reflecting the release of carbon along the flow path have been proposed [Hoque et al., 2012, Mihajlov et al., 2016] considered on the basis of the stable isotopic composition ($^{13}\text{C}/^{12}\text{C}$) of DIC but they are highly uncertain. The average $\delta^{13}\text{C}$ of -4.5‰ ($\sigma = 2.2\%$, $n = 33$) in deep groundwater DIC within the study area shows considerably less depletion than deep groundwater in other parts of the basin, with an average $\delta^{13}\text{C}$ of -16‰, ($\sigma = 6\%$, $n = 39$) (Figure 2c). We do not have sufficient data for the detailed geochemical modeling required for correction in such a case [Aravena et al., 1995] and no further corrections were attempted.”*

Khan et al. (2019) is not the only evidence that we use to prove that these aquifers are Pleistocene in age. Stratigraphy and LGMP provide geological constraints. Multiple references cited in this work infer deep aquifer systems in coastal BLD with similar depth range of our study have Pleistocene in age. We also added the ^{14}C activities from Majumder et al. (2011) in the discussion. Khan et al. (2019) also included groundwater ages from other papers to make comprehensive analysis. Therefore, we used this published data to provide geochemical constraints on groundwater ages in our study.

In the main text, I revised the section using the uncorrected age from Khan et al. (2019). I added more explanations where the Pleistocene groundwater age is applicable. See the revised “Fossil fresh aquifers” section for more details.

Comment 14: *L106-118 preserved Pleistocene groundwater vs. resemblance to groundwater system offshore New Jersey: It should be clarified, if only the morphology or the (emplacement) processes are expected to be similar*

Reply 14: The point when comparing the morphology between these two aquifers is to show that R1 is potentially connected to upgradient recharge (albeit which would be extremely slow recharge due to the low topography gradient) and not isolated pockets. The emplacement processes of the two aquifers might be similar; however, the sediment supplies and geological settings of the two deltas are different. Adding this complexity to the comparison will lose track of interpreting R1. Like our response to direct comment in the main text, we decided to neglect the comparison between the preserved Pleistocene groundwater in coastal BLD with groundwater systems offshore New Jersey to keep track with the length of the paper and not losing focus on the main aquifer R1.

Comment 15: *L127: refer to Fig. 2*

Reply 15: We added Fig. 2 to the end of the sentence.

Comment 16: *L152: Sea-Level Change*

Reply 16: We fixed the typo.

Comment 17: *L302: refer to Fig. 1c*

Reply 17: We referred to Fig. 1c on line 302.

Comment 18: *L456: add link*

Reply 18: We will make all data publicly prior to publication.

Figures and Tables:

Comment 19: *Are Figures 4a and 4c and Table 1 mentioned in the manuscript text?*

Reply 19: We referred figures 4a and 4c to appropriate places in the main text. Previously, we used them together as Fig. 4.

For table 1, I did not mention it in the main text. Table 1 could go together with Fig. 2. Table 1 lists the bulk resistivity and estimated salinity of features in Fig. 2a and 2b. Although the color scales in Fig. 2a and 2b indicate clearly the values, table 1 provides the upper and lower limit of each feature. To save space in the published version, we have eliminated this table.

Comment 20: *L 470: please delete this line*

Reply 20: We deleted this line.

Comment 21: *Fig. 1: legends in a and b are too small, an in b the red entries are difficult to see*

Reply 21: They are elevation with color-coded scale. The resolution of the figure is not great in the word document. We enlarged the text of the legends of Fig. 1a&b in the revised manuscript. We enlarged the legends of Fig. 1a and 1b. We will upload the high-resolution figures with the revised manuscript.

Comment 22: *L475: not clear how subaqueous delta depth an SoNG can be distinguished*

Reply 22: The Swatch of No Ground (SoNG) is color-coded as blue, sea and lake, which has elevation less than 0 m. The depth of SoNG is based on bathymetry. If you think that the information is irrelevant, we can erase it.

Comment 23: *L481 not clear where shrimp farms are (dark blue to purple appears a bit masked)*

Reply 23: The color of shrimp farms (dark blue to purple) is clear due to its high contrast with the pink color of fallow agricultural fields and green color of vegetation. Channels and river mouths have blue colors, but their pattern can be distinguished from shrimp farms. Note that shrimp farms are primarily next to channels.

Point-by-point response to referees' comments

Comment 24: *Fig. 2: The axis gives distances (+/-60 km), where P11 appears to be the center. Why did you choose this scale (and the center at P11) instead of 0 to 120 km? And why are Sundarbans explicitly mentioned here?*

Reply 24: P11 is very close to the center of the profile, but it is not at the center. This scale was used while we did inversion models for resistivity. The default setting of the code chose the center of the inversion profile as the origin. Readjusting the distance scale makes no difference in absolute values. We added the Sundarbans here to help readers know the locations of the groundwater bodies and the saline water gap. Readers can use the map in Fig. 1c and 3 to visualize the location of these structures. We added specific four stations and North-South symbols with the same purpose.

Comment 25: *Fig. 3: connection between Fig. 3 and Fig. 4 somewhat difficult*

Reply 25: Figure 3 shows all available data that we gathered to explain the groundwater distribution in the map view. Figure 4 shows a conceptual model to explain the formation of deep fresh groundwater R1 and R2 due to sea-level fluctuations with time in a cross-section view. The two figures have different purposes. You can use the Ganges paleovalley to connect the two figures in spatiotemporal scale.

Comment 26: *Fig. 4: maybe insert R11 and R2 into Fig. 4c*

Reply 26: We inserted R1 and R2 into Fig. 4c.

Response to Reviewer #3:

Comment 1:

The carbon-14 data used to support the conceptual model lacks geographic specificity, which leads to apparent overinterpretation of the data. The data comes from one study (ref #5, Kahn et al. 2019). Most of the carbon-14 ages in Kahn are Holocene and the subset that are Pleistocene are not identified and located relative to the current study area. Some of the points in Kahn are more than 100km from the closest boundary of the study area. Extrapolating over such a distance is unreasonable in most contexts and certainly in a major delta complex such as the Ganges. Further, the authors argue for the water in R1 and R2 to have been emplaced during the last glacial maximum -- only 7 of the carbon-14 dates in Kahn fall in the range of the LGM, which makes understanding the location of Kahn's data relative to the current study even more crucial.

Confidence in Pleistocene ages for the vast majority of Kahn's samples nearest to the subject study area is key. If they're Holocene, the entire conceptual model falls apart. To invoke Pleistocene emplacement of R1 and/or R2, the authors must unequivocally demonstrate the presence of Pleistocene water in close proximity to their transect.

Reply 1: In Khan et al. (2019), there are two deep groundwater ages: As-affected areas and low-As areas. For the As-affected areas, the younger age of 2.0 +/- 0.6 kyrs might indicate the downward flow of Holocene As-contaminated groundwater rather than the true age of deep groundwater (>150 m). The As-

affected areas do not have the LGMP to prevent vertical mixing and are associated with the Ganges paleovalley areas. For the low-As areas, the older age of 12.0 +/- 4.0 kyrs reflects no or less As contamination. The deep groundwater in these areas is protected by LGMP from vertical mixing or downward flow and is associated with the interfluvial areas. Therefore, they showed groundwater ages of water samples similar to aquifers in our study areas. Although the location of their study is not collocated with our survey area, the two transects in their study go across the Ganges paleovalley located to the northwest of our MT survey. Based on the similarity of lithological setting, we used Khan's study to constrain the imaged fresh aquifers R1 and R2.

In the supplementary materials of Khan et al. (2019), they have a section to explain why they did not use the ^{14}C age correction. The correction applied for the uncorrected ages would reduce the age by about 1.0 ^{14}C kyr, but they would have high uncertainty. Table S1 in Supplementary Materials of Khan et al. (2019) have the corrected ^{14}C age of groundwater for the southern transect. The SWAA40557 sample has an uncorrected ^{14}C age of 29.647 kyrs at 249 m depth and the corrected ^{14}C age of 28.776 kyrs. The KHU40053 sample has an uncorrected ^{14}C age of 15.118 kyrs at 162 m depth and the corrected ^{14}C age of 14.247 kyrs.

We agree with the reviewer that having deep groundwater age collocated with our MT survey will provide better constraints, but currently we do not have groundwater ages of R1 and R2. Thus, we used other evidence to constrain the imaged groundwater R1 and R2 (Fig. 3). The radiocarbon age from Khan et al. (2019) is one of the constraints. Khan et al. (2019) also included groundwater ages from other papers to make comprehensive analysis. Along with the radiocarbon age, stratigraphy and LGMP provide geological constraints. Multiple references cited in this work infer deep aquifer systems in coastal BLD with similar depth range of our study have Pleistocene in age.

We are aware of the potential vertical mixing between shallow Holocene and deeper Pleistocene groundwater due to the absence of LGMP. In these circumstances, deep groundwater may have Holocene ages. For instance, Khan et al. (2019) showed that the ages of groundwater samples located within the paleovalley-areas with no LGMP-the groundwater age is Holocene. Also, the LGMP is laterally discontinuous across GBD that will lead to the variations of fresh groundwater distribution. Thus, our conceptual model still works on a regional scale.

In the main text, we revised the section using the uncorrected age from Khan et al. (2019). We added more explanations where the Pleistocene groundwater age is applicable. See the revised "Fresh fossil aquifers" section for more details.

Comment 2:

In the development of their conceptual model, the authors argue that the fresh water identified in R1 and R2 is relict, connate water emplaced during the LGM, as opposed to the terminal portion of an active groundwater system. At several times they contradict this conceptualization by referencing upgradient connectivity and making comparisons to the Ganges system to an offshore system in New Jersey (specifics noted in comments in PDF file). The contradictory statements in the text undermine their conceptual model. Further, the northern extent of the R1 freshwater region extends an unknown distance outside the study area. There is no evidence presented for the reader not to envision this lobe of freshwater continuing upgradient as part of a continuous, active groundwater flow system.

The deficiencies in this study principally stem from its heavy reliance on data from existing published

Point-by-point response to referees' comments

work in areas that do not exactly overlap with theirs to support their interpretations of their interesting MT transect results. Without site specific data to support their interpretations, the authors often find themselves at odds with the data and with their proposed conceptual model. The original work (the MT transect) appears sound and compelling; its interpretation much less so.

Reply 2: We argued that R1 and R2 are primarily fossil freshwater emplaced during the sea-level lowstand and potentially connected to upgradient recharge. Based on morphology of aquifers, LGMP paleosol map, trace of Ganges paleovalley, R1 and R2 could connect with upgradient recharge sources located over tens to hundreds of kilometers away from location of R1 and R2. Although they might be connected to recharge sources, the recharge rates are extremely low due to low topography and hydraulic gradients. The recharge rates require thousands of years or more. Therefore, these aquifers can be considered as non-renewable water resources. We added references and revised the “Fossil fresh aquifers” section in the main text to clarify our argument. We neglected the New Jersey comparison to focus on the interpretation of R1 (see the responses to these specific comments).

We agree that R1 could extend to the northern part out of the MT profile.

We revised the interpretation based on all the comments. Please check the revised edition for more details.

Again, the radiocarbon age is not the only data we used to interpret our resistivity and salinity models. We also used LGMP outcrop and subcrop as well as Holocene sediment thickness, the trace of paleovalley to explain our geophysical results. We also added the ^{14}C activities from Majumder et al. (2011) in the discussion. The compelling parts of our work are imaging the full depth extent or thickness of groundwater systems rather than the top of the aquifers, the presence of LGMP controlling the freshwater distribution, and the past sea-level cycles contributing to deep groundwater emplacement. The proposed conceptual model helps visualize these key results and dynamic processes involved in deep groundwater emplacement.

Direct Responses to Comments in the Manuscript:

Comment 1: *Line 29-30: reference needed*

Reply 1: We added van Geen et al. (2008) and Rahman et al. (2020) papers.

Comment 2: *Line 50: “S” is capitalized in caption of fig 1*

Reply 2: We changed small “s” to capital “S”.

Comment 3: *Line 53: This colloquial sounding sediment size fraction seems out of place with the specific size-fraction terms used in the rest of the sentence and prior sentence.*

Reply 3: “coarse sand” and “clayey silt” are used to describe variety of grain size: coarse sand (0.5 - 1 mm) to silt (0.05 - 0.002 mm) and clay (<0.002 mm). “Clayey silt” represents major grain size in the Sundarbans mangrove forest, which is silt-dominated mixture with sand.

Comment 4: *Line 55: Not on map. The reader has no context for where these are. Are they important to even mention?*

Point-by-point response to referees' comments

Reply 4: The two terraces are on Fig. 1b. I put "(Fig. 1b)" in the main text to show them. We think it worth to mention them because they have LGMP outcrops compared to LGMP subcrop, which is relevant to Pleistocene fossil groundwater. They could be the location where groundwater recharge occurs for Pleistocene aquifers, although the timescales could be thousand years or more.

Comment 5: *Line 57: consider hyphenating: fresh-to-saltwater*

Reply 5: We added hyphens.

Comment 6: *Line 58-59: The statement about compaction appears unsupported. There are refs before it that seem to document high sedimentation rates and refs at the end of the sentence that seem not to be related to the topic of compaction. Please clarify.*

Is there potential for mass loading (independently of compaction) to contribute to subsidence?

Reply 6: We included the subsidence to provide background information relevant to the study area. The subsidence due to compaction contributes to the depth of Holocene-Pleistocene sediment boundary.

This boundary separates shallow Holocene groundwater, which has multiple problems mentioned above, to Pleistocene groundwater, which is fresh and potentially As-free.

Also, the subsidence creates the accommodation space for sediments and affects the evolution of river systems in the study area.

References at the end:

28. Allison et al. (2003) indicate that the regional subsidence of Bengal Basin affects the evolution of Holocene sediments in the delta.

29. Grall et al. (2018) show the subsidence increase in seaward direction.

30. Steckler et al. (2022) provide quantitative evidence of compaction and subsidence. Subsidence and compaction vary with depth, space, and time.

31. Michels et al. (1998) support delta front seaward progradation. Label F in Fig. 4b demonstrates the delta progradation.

Yes, mass loading plays a role in subsidence. In our study area, young unconsolidated sediments have high porosity. Therefore, adding mass on top of older sediments will cause compaction to occur.

Comment 7: *Line 60: interesting but not related to the rest of the sentence and could be dropped with no loss if you need to save space or expand elsewhere.*

Reply 7: We agree. So, we erased the phrase.

Comment 8: *Line 63: Reference needed*

Reply 8: We added Uimitsu (1993) as reference for Pleistocene sea-level change in coastal BLD.

Point-by-point response to referees' comments

Comment 9: *Line 64: Misleading. This statement appears to be expanding on the statement in the topic sentence, which specifically mentions the GBD. Neither of the two references supporting this statement are from the GBD. Rephrase to clarify this is a generic world-wide statement, or cite supporting documents specific to GBD.*

Reply 9: We changed references from global scale to local GBD by using Uimitsu (1993), Goodbred and Kuehl (2000), and Goodbred et al. (2003).

Comment 10: *Line 101-103: This is speculative and seems plausible, however nothing in the remainder of this section definitely supports the statement.*

It seems the water in both R1 and R2 could be part of modern flow systems that originate upgradient. Specifically, R1 maintains its full thickness all the way to the left (north side) of figure 2. The upper limit and connectivity is unknown. On line 120, the authors acknowledge the seeming disconnection of R2 is due to the geometry of their transect and the water is related to flowpaths originating to the NW of R2 and off the N-S orientation of their MT survey.

Reply 10: The LGMP protecting the upper boundary of two aquifers from vertical mixing is a primary feature to separate R1 and R2 from Holocene (modern) flow systems.

Both R1 and R2 could connect to upgradient recharge systems, but the extremely small gradient of low-lying coastal GBD requires timescales of thousands of years or more for recharge. Wilson and Goodbred (2015) (ref. 48) show that the flatter lower delta plain has gradient of 10^{-5} , which is 1 meter change in slope over 100 km. Thickness of R1 extending towards the northern side at depth means Pleistocene freshwater can extend farther north of the profile. However, the timescales required for recharge in our survey areas is not associated with modern flow systems.

Currently, we do not have isotope data at R1 and R2 to constrain the age and flow systems of the two aquifers. Thus, we gather other information to explain the distribution of R1 and R2.

The Holocene isopach is one of the geological indicators for the age and extension of groundwater. We indicated in Fig. 3 that the Holocene sediment thickness near the vicinity of R1 is 90 m. Therefore, we stated that the aquifers here are likely associated with Pleistocene or older strata. The LGMP created a natural barrier to prevent vertical mixing between modern (Holocene) groundwater with paleowater.

Comment 11: *Line 106: Ref needed.*

Reply 11: We added Sarkar et al. 2009 and Pickering et al. 2014.

Comment 12: *Line 109-111: ref needed.*

Reply 12: We moved the Hoque et al. (2014) and McArthur et al. (2008) to the end of the sentence.

Comment 13: *Line 109: P-M reversed in acronym*

Reply 13: We changed the order.

Comment 14: *Line 111-112: unsupported speculation*

Point-by-point response to referees' comments

Reply 14: The LGMP is the fine-grained transgressive sediments mentioned above. The paleo-fresh groundwater could not vertically mix with Holocene groundwater due to LGMP. I added McArthur et al. (2008).

Comment 15: *Line 112-113: Kahn et al. clearly show the presence of some Pleistocene groundwater GW at 200-300m in the Ganges delta, however most of their dates are Holocene, including a large concentration of dates around 10ka. It is not straightforward to identify the Pleistocene samples in Kahn et al. that are within or reasonably proximal to the present study.*

Because the authors of this study rely on the age dates in Kahn et al. to support their conceptual model (they are key to the assertion that the R1 and R2 are relict Pleistocene water), the onus is on the authors to equivocally demonstrate the Pleistocene samples in Kahn et al. are within the current study area (or meaningfully proximal). It should not be up to the reader to piece this together. If it cannot be demonstrated that there is Pleistocene water in or very near their study area, the conceptual model needs to be revised.

Reply 15: We do not have isotope data collocated with the MT survey profile. Thus, we used Khan et al. (2019) as a reference to provide constraints on groundwater ages. Since their samples cover both the interfluvial and paleochannel areas, we can use them to correlate with LGMP and infer groundwater ages close to our survey. Khan et al. also include other studies to compare and investigate the deep flow system on a regional scale.

Khan et al. (2019) is not the only evidence that we use to prove that these aquifers are Pleistocene in age. Stratigraphy and LGMP provide geological constraints. Multiple references cited in this work infer deep aquifer systems in coastal BLD with similar depth range of our study have Pleistocene in age. Khan et al. (2019) also included groundwater ages from other papers to make comprehensive analysis. Numerical modeling from Michael and Voss (2008) indicates low regional horizontal recharge.

In Khan's paper, there are two deep groundwater ages for As-affected areas and low-As areas.

For the As-affected areas, the younger age (2.0 +/- 0.6 kyrs) might indicate the downward flow of Holocene As-contaminated groundwater rather than the true age of deep groundwater (>150 m).

For the low-As areas, the older age (12.0 +/- 4.0 kyrs) reflects no or less contamination and can be a true age. The groundwater ages of these areas are more relevant to fresh aquifers in our studies since vertical mixing or downward flow is prohibited by LGMP. Mihajlov et al. (2016) show that deep aquifer (>150 m) in central Bengal Basin was last substantially recharged around ~10 kyr ago. The last substantial recharge at ~10 kyr was likely to occur in coastal Bangladesh and might be a reason for a large concentration of dates around 10 ka. Therefore, the true deep groundwater ages must be older, and the 10 ka only shows the last time of recharge.

Most low-As samples in Khan et al. have uncorrected ^{14}C ages spanning within 10-20 kyrs and two samples up to 30 kyrs. In their supplementary materials, they stated that the corrected ^{14}C age may decrease by 1.0 kyrs compared to the uncorrected ages. More correction methods can be applied, but they also have high uncertainty. Table S1 in Supplementary Materials of Khan et al. (2019) have the corrected ^{14}C age of groundwater for the southern transect. The SWAA40557 sample has an uncorrected ^{14}C age of 29.647 kyrs at 249 m depth and the corrected ^{14}C age of 28.776 kyrs. The KHU40053 sample has an uncorrected ^{14}C age of 15.118 kyrs at 162 m depth and the corrected ^{14}C age of 14.247 kyrs.

Point-by-point response to referees' comments

Comment 16: *Line 115: I'm not sure this is a useful comparison. The cited study indicates the Pleistocene water is found in fine-grained units separated by saline water in higher-permeability units.*

The model proposed by the authors herein on lines 101-103 seems to envision a pocket of relict Pleistocene water that stands in isolation to the modern flow system. They imply this relict water is in a permeable unit (otherwise it could not be developed as a drinking water resource). However, the notion of it being connected to modern upgradient systems arises later, including here, by leaning on this NJ study and later on lines 119-121. However, even after recognizing the potential continuity with upgradient systems, a static period of formation during the LGM is described on line 122.

Connection with the upgradient GW system also is indicated by the morphology of R1 at the northern end of the transect -- there is no indication of a terminus and indeed looks very much like the nearshore/offshore terminus of the on-shore deep freshwater GW system described in the NJ article and in many other coastal GW systems.

Consistency and clarity is needed in the authors' conceptualization of the system, and its alignment with existing data in the Ganges delta.

Reply 16: First, we show the similarity in morphology of R1 in this study and New Jersey offshore aquifer to argue that R1 might have upgradient recharge at very low rate due to low topography. Since we do not have a full image of entire aquifer related to R1-reviewer points out that R1 maintains its thickness to the north-using this similarity might help clarify the point. Isolated pockets of freshwater will have different shapes compared to the connected ones.

In New Jersey system, the freshwater stored in Miocene fine-grained sediments. We are aware of the complexity and differences between two delta systems in terms of stratigraphy, sediment supplies, and groundwater ages. That's why we only consider the shape of groundwater bodies.

To keep track of the length of the paper and not losing focus on the main aquifer R1, we neglect the comparison with the New Jersey system.

Comment 17: *Line 122: This is a significant overinterpretation of the age dates in Kahn et al that are being used to support a Pleistocene water source. Taking 20-26ka as the duration of the LGM, only 7 of the wells in Kahn's dataset correspond to LGM and the location of those 7 wells is not provided to the reader of this article. As noted earlier, the onus is on the authors here to demonstrate those LGM samples are spatially relevant to their arguments.*

Reply 17: We agree that we have minor constraints on R2 in terms of groundwater ages and LGMP extent. We changed the suggested timing of formation prior to the Holocene due to the dimensions of R2 and stratigraphy. It is likely to have Pleistocene in age, but no further constraints on that.

Comment 18: *Line 127-129: The traverse and paleochannel identified by Hoque is considerably north of the end of MT transect herein. Their transect is about 23.3° latitude and the proposed paleochannel location shown on figure 3 herein is at about 22.3° latitude.*

Perhaps reference 43 (Sincavage et al. 2017) provides information on a more southerly channel but I was unable to access the article.

Reply 18: We traced the Ganges paleochannel further down to Sundarbans area in Fig.3 using its thalwegs. The transect of the Hoque et al. (2014) cover the interfluvial and paleochannel areas. We had no

Point-by-point response to referees' comments

problem with accessing Sincavage et al. (2017), so we can share this paper with the reviewer if necessary. The study area of Sincavage et al. (2017) is in Sylhet, not near the southerly channel.

Comment 19: *Line 133-137: Speculative*

Reply 19: Yes, we clarify that we speculated this idea.

Comment 20: *Line 148-151: Assigning an early Pleistocene data based on imaging alone is highly speculative. Why could it have not occurred during a subsequent interglacial or even during the earlier Pliocene? No data is provided to constrain. Even if the interpretation of R1 and R2 is correct, a LGM date for them does not provide any sort of constraint on how long before their emplacement the saline water in C2 must have occurred.*

Reply 20: We revised the hypothesis of the emplacement of conductor C2. However, we do not have direct constraints from boreholes, geophysical or geochemical data for C2. Therefore, we used offshore Sangu-1 well and seismic data to constrain C2.

We agree that we do not have any absolute age constraint for the marine deposition. Thus, we change to relative timing of events based on spatial relationship between C2 and R1 & R2.

Comment 21: *Line 155-157: neither statement is satisfactorily substantiated*

Reply 21: we added a combination of evidence to clarify the statement. More details are in the “Fossil fresh aquifers” and “High salinity zones” sections.

Comment 22: *Line 289: I read through this but am not a geophysicist and the content was outside my area of expertise to critique.*

Reply 22:

Comment 23: *Line 386-387: Perhaps I misunderstand the context of the term “geothermal gradient” as used here. As I have experience it’s use, it refers to the increase in temperature with depth due to conductive heat movement out of the earth. In this context of this sentence, it is implied that the geothermal gradient is somehow related to air temperature.*

Reply 23: The geothermal gradient here accounts for temperature increase with depth. Groundwater temperature is potentially higher at depth due to heat transfer from surroundings. Therefore, the temperature of groundwater is higher than the surface water temperature, which is between 25 and 28 degrees Celsius. Adding geothermal gradient factor to the conversion prevents overestimating fluid salinity at deeper depth.

Comment 24: *Line 389-390: This would seem to pose a problem for the presence of low-permeability units that have been proposed and invoked as significant impediments to flux of Pleistocene water out of zones R1 and R2.*

Reply 24: The proportion of sand is higher in Pleistocene sediments than in Holocene unit in coastal BLD should not pose a problem for pumping groundwater out of the Pleistocene aquifers. Multiple deep wells have been drilled to support local communities near Khulna city and elsewhere.

Point-by-point response to referees' comments

Comment 25: *Line 390-391: unusual phrasing*

Reply 25: We changed “dominates” to “is higher than”

This phrase describes the conductivity effect of saltwater is the first-order control of the bulk conductivity. The conductivity effect of clay is lower than that of saltwater. Therefore, Archie's law can be applied to estimate fluid resistivity/conductivity without violating its assumption.

Comment 26: *Line 390: “deeper” would perhaps be a bit more precise*

Reply 26: We changed as suggestion.

Comment 27: *Line 394: provided*

Reply 27: We fixed it.

Comment 28: *Line 492: define term for those unfamiliar*

Reply 28: We added text next to PSS78 to clarify.

Comment 29: *Line 495: Please make clear if this comment refers to both figures 2a and 2b. If only 2b, please clarify which two scales it refers to as I interpret it as having just one scale ranging from yellow to blue.*

Reply 29: We added text next to PSS78 to clarify.

Date: June 28, 2025

Manuscript Number:

Title of Article: **Hidden deep freshwater reserves beneath salinity-stressed coastal Bangladesh**

Name of the Corresponding Author: Huy Le

Email Address of the Corresponding Author: hdl2115@columbia.edu

Dear All,

We would like to thank the two reviewers for their second round of feedback and the editor for the time extension to edit our manuscript. We have incorporated the reviewer's comments to revise the discussion of deep freshwater systems in coastal Bangladesh. Below we provide the reviewer comments in black text and our replies in blue.

Once again, thank you all for your consideration to our manuscript.

Best regards,

Huy Le and coauthors

MAJOR REVISION

- Modify the interpretation of fossil freshwater to partial recharge freshwater reserve for R1
- Clarify the ^{14}C age profile in Figure 3 and re-interpret groundwater age from Khan et al. (2019) to explain partial recharge of R1.
- Expand the significance of this study in using MT method to find deep freshwater resources and the widespread presence of paleo-valleys during the sea-level lowstand.
- Modify the title, abstract, subtitles, and discussion due to the change in R1 interpretation.

REVIEWER COMMENTS

Reviewer #2 (Remarks to the Author):

I thank the authors for thoroughly revising the manuscript. It has significantly improved, and many aspects have been clarified. Also, comments have been addressed in detail. Yet, though in discussions absolute age derivations have been largely changed to relative timing where direct constraints were missing, regarding uncorrected ^{14}C ages I still would suggest to incorporate some (or a concise version) of the explanation into the manuscript or supplement that the authors provided in their reply to Comment 13 (reviewer 2) and "Comment 15 of the Direct Responses to Comments in the manuscript", especially regarding the Khan et al. study. Also, highlighting the uniqueness of their study as suggested in the first revision was moderately addressed, which might still be slightly improved. As a minor point I suggest to check for spelling/typos (especially the revised passages).

Reply: We thank reviewer 2 for addressing the remaining issues of our revised manuscript.

The uncorrected 14C ages from Khan et al. (2019):

Line 139-151: We modified our interpretation from Khan et al. (2019) groundwater data. Given the uncertainty of the uncorrected ages, we interpret the freshwater body R1 contains both Holocene and Pleistocene groundwater since the 14C age data within the interfluvial cluster around 10-12 kyrs with two low-As wells exhibiting Pleistocene groundwater ages. Therefore, we interpret R1 as a freshwater reserve containing paleowater with slow regional recharge instead of primary fossil freshwater. We expanded the interpretation in this second revision compared to the previous version but keep the main text to be succinct. Thus, readers can check the original paper for the 14C age correction. We added the deep groundwater age samples from Khan et al. (2019) to Figure 3 in the Main Text for clarifying their location relative to our MT survey. I hope that the revised explanation addresses the remaining concern of reviewer 2.

The uniqueness of this study:

In the last section “Sea-level change and the formation of deep freshwater resources”, we emphasize the role of sea-level fluctuations and sedimentary processes in deep groundwater formation.

Line 203-207: we added the significance of incised paleovalley controlling the freshwater distribution. Incised paleovalley is a widespread phenomenon during sea-level lowstands in Pleistocene. Paleo-river could erode soft, unconsolidated sedimentary layers and remove the impermeable layer to confine deep groundwater system. We added three references associated with paleovalley (refs 51, 60, 61). We also clarified that paleovalley can act as conduits for deep groundwater salinization.

51. Larsen, F., Tran, L. V., Van Hoang, H., Tran, L. T., Christiansen, A. V., & Pham, N. Q. Groundwater salinity influenced by Holocene seawater trapped in incised valleys in the Red River delta plain. *Nature Geoscience*, 10(5), 376-381. (2017).

60. Meyer, R., Engesgaard, P., & Sonnenborg, T. O. Origin and dynamics of saltwater intrusion in a regional aquifer: Combining 3-D saltwater modeling with geophysical and geochemical data. *Water Resources Research*, 55(3), 1792-1813. (2019).

61. Ta, T. K. O. et al. Latest Pleistocene to Holocene stratigraphic record and evolution of the Paleo-Mekong incised valley, Vietnam. *Marine Geology*, 433, 106406. (2021).

Line 213-219: We emphasized the significance of using MT method to capture the full dimensions of deep groundwater for volumetric estimation, compared to other EM techniques.

We double-checked the spelling and typos in the second revised manuscript.

Point-by-point response to referees' comments – Second Revision

Reviewer #3 (Remarks to the Author):

This is a review of the revised manuscript titled "Deep fossil freshwater in coastal Bangladesh" by Huy Le and others. The revised manuscript largely addressed comments I had and those of the additional reviewers. I appreciate the authors' improvements to the manuscript, in particular the "Fossil Fresh Aquifers" section. This section provides better background for someone not familiar with the impressive body of work on the GBD and more cohesive support for the authors' interpretations (despite the few remaining concerns about the emplacement mechanism and age of R1 discussed below).

I reread and reviewed the entire manuscript for technical content but stopped commenting on editorial issues after the first 50 lines or so. There were several instances of either awkward phrasing or missing words. An editorial review prior to resubmission is recommended. The following items were noted for consideration by the authors.

L22

consider adding "the" before "location"

L34-35

consider "...deep groundwater at a regional scale..."

L35

consider changing "sustainability" to "susceptibility"

L69-70

The topic sentence on L64 limits important sea-level fluctuations to the Pleistocene

L108-193

I can accept R2 being relict Pleistocene water as it appears to be cut off from the modern flow system by the upgradient paleochannel and is more saline than R1. I have more difficulty with the certainty expressed in the age of R1 and its origin ascribed to a body of water "formed" during the Pleistocene. I think the issue is largely semantics.

Firstly, the origin of the water. The language used in the text implies connate water, although that term is no longer used in the manuscript, e.g:

"freshwater bodies formed during the Last Glacial Maximum's (LGM) sea-level lowstand and subsequently sealed by a finer-grained, low-permeability layer" [L108-110]

"remnants of preserved Pleistocene groundwater" [L180]

Intentionally or not, the language implies the water in R1 originated in that location and has remained there for 12,000+ years. Perhaps it has, however, that has not been demonstrated nor is it not supported by subsequent statements that indicate R1 is the distal end of a larger upgradient flow system (L123-125) with slow transport times (L123-125).

Secondly, the timing of recharge of the groundwater in R1. The location of the 14C ages supporting the assertion of Pleistocene water is provided only as "northwest of our profile". Other than the fact those samples are somewhere in the western portion of the 100,000 km² GBD (L50), the reader has no information whether these data are relevant -- were they collected 1 km, 10 km, or 100 km away? There is a significantly stronger case for transferability if they're nearby. Consider showing the locations on Figure 3 if they are within the map extent. The transport time of "thousands of years or more" (L124-125) is consistent with Holocene as well as Pleistocene recharge. The 14C data presented provide evidence for shallow-deep difference in age in another part of the GBD system having similar characteristics to the region of R1, which is useful, but not conclusive given the lack of information about their location. The evidence presented for Pleistocene water in R1 is not sufficiently compelling to make a definitive assertion as to its age, particularly in light of the wide range and complex distribution of groundwater ages presented by Kahn et al. (2019), from which the example 14C data were taken.

Consider using the uncertainties surrounding the emplacement mechanism and timing of R1 in your favor instead of trying to force certainty with imperfect data. For example, if R1 is the distal end of a long, slow flow system, it is potentially a more usable and renewable resource than connate Pleistocene water. If the age of R1 is mid-Holocene (for example) rather than Pleistocene, it could indicate a more useful/renewable source. Since the groundwater system hosting the freshwater in R1 is relatively coarse-grained (capped by the clayey paleosol), could you pump water from R1 without substantial drawdown? Could you potentially induce a bit more recharge from modern recharge zones or set up managed aquifer recharge systems? The MT transect is a really useful first step to asking the next set of important questions relating to development and management of the freshwater resource in the GBD

Reply: We thank reviewer 3 for the thorough feedback on our revised manuscript. For this new revised version, we modified and added details to address the remaining concerns.

L22

consider adding "the" before "location"

L34-35

consider "...deep groundwater at a regional scale..."

L35

consider changing "sustainability" to "susceptibility"

L69-70

The topic sentence on L64 limits important sea-level fluctuations to the Pleistocene

We changed the terms based on these recommendations.

Line 108-193:

Again, we thank reviewer 3 for the suggestions to improve our interpretation. See “Freshwater reserves” section for more details. Here, we summarize the main change based on your feedback:

We added the groundwater age samples from Khan et al. (2019) to Figure 3 in our manuscript to show the location of their data relative to our MT survey.

We agree that the groundwater ages show both Holocene and Pleistocene ages. Therefore, we modified our interpretation for R1 from primary fossil groundwater to a freshwater reserve containing both Holocene and older groundwater due to regional recharge.

We reinterpreted Khan et al. (2019) data in lines 139-151. Given the uncertainties about the timing of R1 and other potential emplacement mechanism, we propose that R1 is likely to form during the LGM sea-level lowstand with slow regional recharge during the Holocene.

We also checked words and contexts that imply fossil freshwater in the text to remove the ambiguity of timing since MT only provides mainly spatial constraints, and we used other studies to put temporal constraints on our results.

We also proposed two hypotheses of R2 formation. It could be a relict paleo-groundwater due to R2's isolation from R1 and northern recharge. An alternative explanation is that R2, like R1, is a freshwater reserve with slow regional recharge from West Bengal. However, we had limited constraints on R2.

We also checked typos and weird phasing in our main text.

Date: October 2, 2025

Manuscript Number: **NCOMMS-24-67705B**

Title of Article: **Buried deep freshwater reserves beneath salinity-stressed coastal Bangladesh**

Name of the Corresponding Author: Huy Le

Email Address of the Corresponding Author: hdl2115@columbia.edu

Dear All,

We thank two reviewers for their approval of the previous version of our manuscript and suggestions to complete the final draft. We have incorporated these changes to the main text. Also, we provide the reviewer comments below in black text, and our replies in blue.

Best regards,

Huy Le and coauthors

REVIEWER COMMENTS

Reviewer #2 (Remarks to the Author):

I thank the authors for their revisions. My comments have been largely addressed. An aspect that is not clear (and which relates to my former review suggestions regarding 14C) can still be found in lines 150 to 152, where the authors state that „Although uncorrected 14C ages are reported instead of corrected ages, which decrease the uncorrected values by 1.0 14C kyr, due to the high uncertainty associated with initial 14C content corrections, ..“ This reads as if all uncorrected ages can be corrected by 1.0 kyr, which does not appear to be the case. I assume the authors intended to state something else (– and certainly not relate to all samples). And if the „1.0 kyr“ is meant to relate to specific samples/locations, some more information or context would be helpful (and it would be interesting to know why corrected 14C can be given in this case and not elsewhere in the study area, in case this is meant here).I would hence would suggest to rephrase this sentence, and also to cut this (5 lines) long sentence into separate sentences.

Reply: We modified the sentence to clarify the point from lines 150 to 152.

Since the original work from Khan et al. (2019) only reported the uncorrected ages for all samples in their main paper, they reported the corrections in their supplementary materials and explained why the correction values were not used. However, the high uncertainty in initial ¹⁴C content corrections does not change the primary findings in their paper that low-As groundwater are associated with deep aquifers with Pleistocene in age. Therefore, we clarified in the previous version that we used the uncorrected values to interpret the groundwater ages and added details to explain why using these values is still applicable. We summarized their work below:

“Although uncorrected ^{14}C ages are reported instead of corrected ages, which decrease the uncorrected values by 1.0 ^{14}C kyr, due to the high uncertainty associated with initial ^{14}C content corrections¹³, the groundwater ^{14}C data exhibiting both Holocene and Pleistocene ages...”

In the modified version, we cut off the confusing part “which decrease the uncorrected values by 1.0 ^{14}C kyr” and separated the long sentence into two sentences. We focused on the main observations and interpretation here and not delved into details about the corrections or groundwater sample analysis in Khan et al. (2019).

Fixed:

Line 148-152:

“We reported the published uncorrected ^{14}C ages instead of corrected ages because of the high uncertainty in correcting for the initial ^{14}C content¹³, however, these ^{14}C ages delineate a clear spatial pattern in the distribution of deep groundwater ages. The older groundwater in interfluvial areas suggests that deep confined systems protected by the LGMP, like the R1 reservoir, are likely to preserve paleowater at least from latest Pleistocene with limited slow regional recharge during the Holocene.”

Furthermore, I would suggest to clarify line 163 where “... the LGMP surface is not mapped“, whether it does not extend as far, or if it could not be mapped (for technical reasons), as in the current form the relation to “..reducing its isolation..“ (L162) does not appear straightforward.

Reply: Thank you for your suggestion. We clarified the southernmost extension of the LGMP.

Line 163-164:

“Unlike R1, the LGMP surface is not mapped as far south as R2, reducing its isolation from saline Holocene groundwater and regional mixing.”

Fixed:

Line 161-164:

“The LGMP surface may end or become deeper than the maximum depth of tube wells at the southern limit of data⁴⁴. In the Sundarban Mangrove Forest, the LGMP cannot be mapped due to the lack of wells. If the LGMP did not develop as far south as R2, it would reduce R2's isolation from saline Holocene groundwater and regional mixing.”

Reviewer #3 (Remarks to the Author):

The authors have provided an excellent revision to their manuscript. The arguments and conclusions are considerably stronger and make for a compelling interpretation of their excellent MT transect. I offer only

a handful of very minor suggestions for clarification. Best wishes for successful publication!

L81 A reference to Figure 2b would be helpful after "...southward dip in its upper boundary." I found myself looking at figure 2a and trying to understand how the authors arrived at an aquifer depth of 600 m, which is clearly shown in Figure 2b. Alternatively, if it is desired to stay with the resistivity measurements, consider a different depth reference that coincides with one of the contour intervals on Figure 2a.

Reply: Thank you for your clarification.

We agree that Figure 2b shows the depth extension of R1 up to 600 m. Initially, we used 3 Ω m contour to describe the resistivity anomalies, R1 and R2. After doing salinity conversion, we followed the 3 psu contour in salinity (Fig. 2b) to interpret depth extension of R1 in the main text. This explains why we used the "600-m" for depth of R1. To keep the description of dimensions of R1 and R2 consistent with resistivity contours in Fig. 2a, we modified the "600-m depth" to "800-m depth" on line 78.

L113-117 It seems as if two different mechanisms are provided to explain the capping of the freshwater sand units. The first sentence invokes rapid deposition during transgression, while the second sentence invokes a paleosol. Please clarify.

Reply: We clarify the mechanism protecting deeper freshwater from vertical mixing with shallow saline groundwater. The LGMP formed during the glacial period is the primary factor protecting the freshwater layer.

Line 113-117:

"A subsequent rapid sea-level rise flooded the coastal GBD^{45,48}, trapping freshwater beneath fine-grained transgressive sediments^{31,44,48}. The Last Glacial Maximum Paleosol (LGMP)⁴⁴, a stiff clay layer formed by weathering of the Pleistocene interfluvial deposits exposed during glacial periods (Ext. Fig. 7), acts as a barrier inhibiting vertical mixing between dense saltwater and lighter freshwater below^{44,48}."

Fixed:

Line 111-115:

Weathering of the Pleistocene interfluvial deposits exposed during glacial periods (Ext. Fig. 7) created a stiff clay layer known as the Last Glacial Maximum Paleosol (LGMP)⁴⁴. The subsequent rapid sea-level rise flooded the coastal GBD^{45,48}, trapping freshwater beneath fine-grained transgressive sediments and the LGMP^{31,44,48}. The LGMP acts as a barrier inhibiting vertical mixing between dense saltwater and lighter freshwater below^{44,48}.

L554 No location provided for data

Reply: Thank you for pointing that out. We published the data on Zenodo.org website at <https://zenodo.org/records/17353663>. We clarify the data location in Data Availability Section.